# Hot Is Rich—An Enormous Diversity of Simple Trichal Cyanobacteria from Yellowstone Hot Springs

**Jan Kaštovský** [1,*] , **Jeffrey R. Johansen** [1,2] , **Radka Hauerová** [1] **and Mildred U. Akagha** [2]

1   Faculty of Science, University of South Bohemia, Branišovská 1645/31a,
    37005 České Budějovice, Czech Republic; johansen@jcu.edu (J.R.J.); radka.hauerova@gmail.com (R.H.)
2   Department of Biology, John Carroll University, John Carroll Boulevard 1,
    University Heights, OH 44118, USA; makagha22@jcu.edu
*   Correspondence: hanys@prf.jcu.cz; Tel.: +420-387772252

**Abstract:** In 2019, cyanobacterial samples were collected from thermal sites in Yellowstone National Park. In this paper, we discuss the considerable richness of representatives of simple filamentous cyanobacteria. Using a polyphasic approach, one new genus, *Copelandiella*, with two species (*C. yellowstonensis* and *C. thermalis*) and eight additional new species *Albertania prattii*, *K. anagnostidisii*, *Kovacikia brockii*, *Leptolyngbya tildenae*, *L. vaporiphila*, *Nodosilinea calida*, *N. igneolacustris*, and *Oculatella castenholzii* were described. At the same time, by analyzing our new molecular data, we concluded that other genera of trichal cyanobacteria can be merged with existing genera: species within the genus *Plectolyngbya* are herein transferred to *Leptolyngbya*, and if some nomenclatural issues are resolved, species within the genus *Leptothermofonsia* will be subsequently transferred to the genus *Kovacikia*, an earlier synonym.

**Keywords:** thermal; Cyanophyta; blue-green algae; taxonomy; new species; new genus; merging





## 1. Introduction

The fascination with hot springs has accompanied mankind throughout its history, formerly more as a gateway to the underworld, while today we hope more for the medicinal effects of thermal mineral-rich waters. Autotrophic micro-organisms, cyanobacteria and algae, form masses in these places, highly visible to the naked eye. Not surprisingly, they have been studied by phycologists for a very long time—the first thermophilic species, *Mastigocladus laminosus*, was described by Ferdinand Cohn in 1862 from Karlovy Vary Spa [1].

One of the world's most famous thermal sites is definitely Yellowstone National Park. The study of extremophile microorganisms from this area has a particularly rich history. Its cyanobacterial flora was first systematically treated in Copeland's monograph [2]. This publication can now be used as a tourist guide for the algologist—many of the specific species of cyanobacteria that Copeland described (e.g., *Cyanobacterium minervae*, *Thermostichus vescus*, and *Colteronema funebre*) still live in the same springs from which they were described by Copeland.

This impressive work has been followed by many now classic studies on thermophilic organisms [3–5]. With the development of modern methods, the importance of studying these microorganisms is certainly not decreasing. After all, the keystone of molecular biology, polymerase chain reaction (PCR) [6] would not be possible without Taq polymerase; a heat-resistant DNA polymerase isolated from the bacterium *Thermus aquaticus*, which was discovered in 1966 in the Lower Geyser Basin [7].

These sites are very attractive for research; therefore, the total number of publications dealing with these thermal springs and their biological recovery is very high. In the last

decade, there has been a noticeable trend toward investigating ecophysiology in combination with the analysis of genomic data [8,9] and, in particular, the effect of temperature on community composition using modern methods [10–14].

Although the biogeographic and speciation issues of thermophilic cyanobacteria are of extraordinary interest, they have been the focus of only a minority of modern studies [15,16]. Understandably, species surviving the highest temperatures, such as "*Synechococcus* sp.", have received the most attention [8–10,14–16]. Studies on simple filamentous cyanobacteria (usually referred to as *Leptolyngbya* spp.) are considerably less common [17]. This study will examine the phylogeny and taxonomy of several simple filamentous cyanobacteria isolated from diverse thermal springs located in Yellowstone National Park. The taxa we recover are either from well-known described genera such as *Leptolyngbya*, *Kovacikia*, *Nodosilinea*, and *Oculatella*, the sparsely known genus *Albertania*, or from *Copelandiella* gen. nov.

## 2. Materials and Methods

Isolation and characterization of strains

Samples were collected from hot springs and lakes in the western part of Yellowstone National Park (Permit No. YELL-2019-SCI-8134) on 13–23 September 2019. A total of 114 samples were examined on site from localities Ojo Caliente (44.5629292 N, 110.8388147 W), Mound Springs (44.5648956 N, 110.8599667 W and 44.5685722 N, 110.8635394 W), Queen's Laundry Spring (44.5634606 N, 110.8701564 W), Meadow Spring (44.5508444 N, 110.8565683 W), Fairy Meadows (44.5259564 N, 110.8556822 W), Young Hopeful Geyser (44.5439672 N, 110.7845056 W), Firehole Lake (44.5440433 N, 110.7865600 W), Angel Terrace (44.9653947 N, 110.7091997 W), Minerva Terrace (44.9706436 N, 110.7052433 W), Clearwater springs (44.7886475 N, 110.7392136 W), and Black Warrior Lake (44.5444544 N, 110.7864528 W) (Figure 1). Most sites fall into the "neutral-chloride" type of spring (high concentrations of carbonate, chloride and silica); Angel and Minerva Terrace are travertine hot springs (high content of sulfate, bicarbonate, calcium and magnesium, low content of silica) [18].

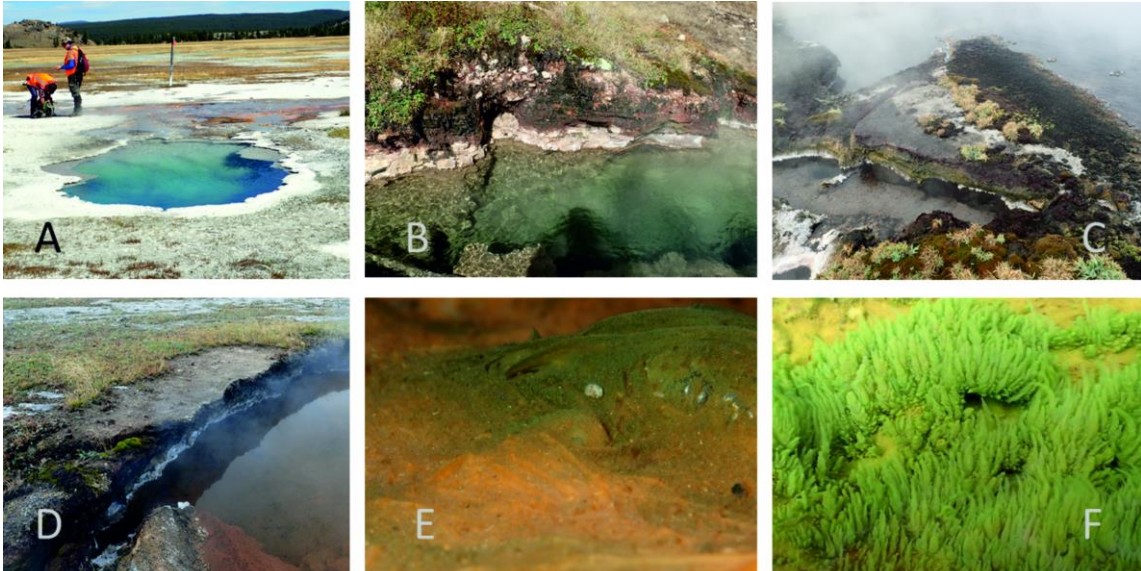

**Figure 1.** Sampling site documentation. (**A**). Fairy Meadow Springs, locality of *Copelandiella yellowstonensis*. (**B**). Clearwater springs, atmophytic locality of *Leptolyngbya vaporiphila* and *Kovacikia anagnostidisii*. (**C**). Black Warrior Lake (*Copelandiella yellowstonensis*, *Copelandiella thermalis*). (**D**). Firehole Lake, atmophytic locality of *Kovacikia brockii*, *Leptolyngbya tildenae*, *Albertania pratii*, *Oculatella castenholzii*, *Nodosilinea calida*, *Nodosilinea igneolacustris*. (**E**). soft mud with blue green mats on bottom of Firehole Lake (*Copelandiella thermalis*). (**F**). Black Warrior Lake, "green octopus" massive periphytic forms from several cyanobacterial taxa, such as *Cyanobacterium minervae*, *Cyanobium amethystinum*, *Heteroleibleinia kuetzingii*, and *Nodosilinea igneolacustris*.

Twenty-seven of the most promising samples were transported to laboratory conditions for strain isolation and cultivation. Solidified Z-8 medium [19] or BG11 medium [20] were used to make enrichment cultures on petri dishes under a 12/12 light/dark cycle and temperature 23 °C. Isolates were picked after 4–6 weeks and put into liquid media. They were subsequently transferred and maintained on solidified media in test tubes under the growth conditions stated above. Most of the strains obtained, thirty-five, belong to filamentous Synechococcales (according to [21], now in Leptolyngbyales, Oculatellales and Nodosilineales according to [22]), so we focused our primary attention on this group. Morphology of strains was studied with light microscopes (Olympus BX53 with camera DP 74 and software cellSens Standard ver. 3.2. ©Olympus Corporation and Olympus BH2 with DP25 camera and cellSense Standard ver. 3.2 ©Olympus Corporation, Tokyo, Japan). Both microscopes utilized Nomarski Differential Interference Contrast (DIC) optics, and images were captured at 1000× magnification. Nomenclature follows Algaebase [23].

Molecular methods

Genomic DNA was extracted from selected strains located in the Cyanobacterial Culture Collection of John Carroll University (JCU-USA) or CCALA at Institute of Botany, Třeboň, Czech Republic, using a Qiagen DNeasy PowerSoil Pro Kit (Venlo, The Netherlands) or CTAB (cetyl trimethylammonium bromide)-based extraction following [24] at University of South Bohemia (JCU-CZ) in České Budějovice, Czech Republic. PCR amplification of the 16S rRNA gene was performed using primers VRF1: 5′-CTC TGT GTG CCT AGG TAT CC-3′ [25,26], and VRF2: 5′-GGG GAA TTT TCC GCA ATG GG-3′ [27,28] or 16S27F-AGAGTTTGATCCTGGCTCAG together with 23S30R-AGAGTTTGATCCTGGCTCAG [29]. All PCR reactions contained 1X GoTaq® Flexi Buffer, 0.025 units/μL GoTaq® Flexi DNA Polymerase, 3 mM MgCl$_2$ (Promega, Madison, WI, USA), 0.2 mM dNTPs, 0.5 μg/μL of BSA (NEB, Ipswich, MA, USA), and 0.5 μM each of primer VRF1 and VRF2, (NEB, Ipswich, MA, USA). Reactions were performed in a BioRad PCR Thermocycler (Bio-Rad Laboratories, Inc., Prague, Czech Republic) with a 3 min incubation at 94 °C to minimize non-specific DNA amplifications. Subsequently, reactions underwent 35 cycles of 94 °C (30 s), 53 °C (30 s) and 72 °C (60 s), followed by an extended incubation at 72 °C (300 s) to complete synthesis. The final dwell was at 4 °C until the PCR products were moved to −20 °C freezer. For the 16S rRNA gene, which occurs as multiple copies across genomes, PCR products were cloned using a StrataClone PCR cloning kit according to manufacturer instructions (La Jolla, CA, USA). Plasmid purification proceeded with a QIAPrep Miniprep Spin Kit (Qiagen, Carlsbad, CA, USA), prior to EcoRI digestion to select successful clones. For each strain, 3–4 plasmids were sent out for sequencing. In the USA, plasmid DNA and purified PCR products were sent to Functional Biosciences, Inc. (Madison, WI, USA) for sequencing, and processed with Sequencher v. 4.10.1 software (Gene Codes Corp, Ann Arbor, MI, USA.). In the Czech Republic, the sequencing was performed by SEQme s.r.o. (Dobříš, Czech Republic) and the sequences were processed in Geneious Prime 2021.0.3 (https://www.geneious.com (accessed on 28 April 2023)).

Phylogenetic Analyses

After sequencing, all clones of each strain were inspected for number of tRNAs in the ITS region. Only operons with 2 tRNA genes were chosen, to reduce errors introduced by sequence differences between multiple paralagous operons. Orthologous operons were identified as those that had identical or near identical (allowing for PCR error) sequences in the ITS region. When we had three cloned orthologues sequences for a strain, consensus sequences were obtained by aligning the three sequences using ClustalW to create one consensus sequence per strain. The 16S rRNA sequences were then aligned with other sequences of simple trichal strains chosen from NCBI Genbank, producing an alignment of 372 sequences (accession numbers for all sequences in Figshare, https://doi.org/10.6084/m9.figshare.23735901.v1 (accessed on 24 July 2023)). In addition, we also looked into the conserved regions of the 16S that have secondary structure (identified by [28]), to make sure that they were aligned correctly, i.e., different nucleotide sequences fold into the same secondary structures. The alignment was submitted to MrBayes on XSEDE (3.2.6) available

on CIPRES Science Gateway v.3.1 [30] using the GTR + G + I evolutionary model for both Bayesian inference (BI) and maximum likelihood (ML) analyses. The BI analysis had a mean estimated sample size (ESS) exceeding 150 for all parameters (ranging 155–2971), above the average of 100 typically accepted as sufficient by phylogeneticists [31]. The final average standard deviation of split frequencies was =0.01. The potential scale reduction factor (PSRF) value for all the estimated parameters in the Bayesian analysis was 1.00, indicating that convergence of the MCMC chains was statistically achieved [32]. Maximum likelihood bootstrap values were mapped onto nodes in the BI analysis. The tree was then broken into three figures, one for each order in which the taxa occur, and many of the nodes were collapsed. The fully intact uncollapsed tree is available in the Supplemental Materials (Figure S1).

The 16S–23S ITS sequences of closely related species clusters were aligned utilizing the secondary structure together with ClustalW. These separate alignments were used to run maximum parsimony (MP) and Bayesian inference analyses, to examine relationships within selected genera. Maximum parsimony analyses utilizing heuristic searches in PAUP were run with the options GAP = NEWSTATE, NREPS = 10,000, MULTREES = YES, SWAP = TBR, and STEEPEST = NO. To determine nodal support, NREPS = 10,000 was used. The BI analysis used the GTR + G + I evolutionary model with two data partitions, one for DNA sequence data, the other for coding indels as standard data (1 = any nucleotide, 0 = indel, and ? = missing data). Bootstrap values from the MP analyses were mapped onto nodes in the BI tree. All trees were post-edited in Adobe Illustrator in CS5.

Alignments for the 16S rRNA and 16S–23S ITS were used to determine percent similarity and percent dissimilarity, respectively. Secondary structures including D1-D1′, Box B, V2, and V3 helices were identified and predicted using the Mfold web server [33]. Additional conserved domains (all helices plus D2, D3, Box A, D4, and D5) were identified for comparison of lengths. All structures were redrawn in Adobe Illustrator in the CS5 software package (Adobe Systems Incorporated, San Jose, CA, USA). Descriptions of secondary structures were based on the nomenclature set forth by [34].

## 3. Results

We were able to isolate over 30 cyanobacterial strains from Yellowstone National Park, most of which represented thin filamentous non-heterocytous forms formerly in the Synechococcocales, as defined in [21], but which are now recognized as belonging to three different orders, the Leptolyngbyales, Oculatellales, and Nodosilineales [22]. Some cultures were lost subsequent to sequencing, but appear in our phylogeny, which contains representatives from all three orders listed, as well as outgroup taxa from the Synechococcales, Prochlorotrichales, Acaryochloridales, Pseudanabaenales, Thermostichales, and Gloeobacteriales. Ten species are recognized, two in the new genus Copelandiella, the rest in pre-existing genera, including *Kovacikia* Miscoe, Pietrasiak & J.R.Johansen [35], *Leptolyngbya* Anagnostidis & Komárek [36] (Figure 2), *Albertania* Zammit [37], *Oculatella* Zammit, Billi & Albertano [38] (Figure 3), and *Nodosilinea* Perkerson Casamatta [39] (Figure 4). These species are almost all morphologically cryptic, in genera that are morphologically similar. However, they are phylogenetically distinct and evolutionarily separated based on both 16S rRNA similarity data and 16S–23S ITS dissimilarity thresholds. They are additionally ecologically separable from other species in their genera, due to their presence in thermal waters, and biogeographically separate from these other species as well. Evidence for lineage separation for all ten taxa will be given in the descriptions of each species, while a discussion of these mostly recently described genera will follow in the discussion.

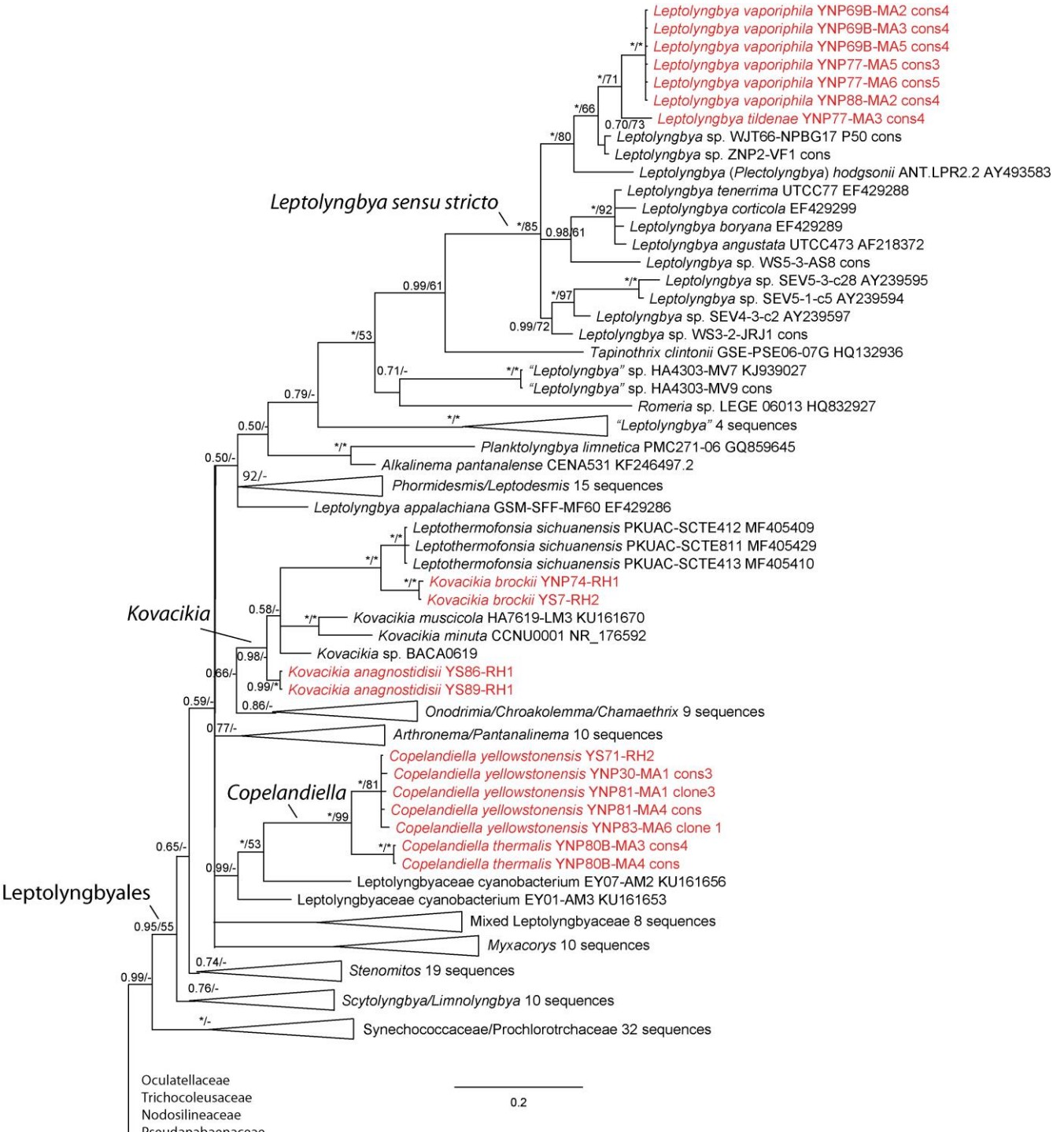

**Figure 2.** Bayesian inference analysis based on 16S rRNA gene sequence with maximum likelihood bootstrap values mapped to nodes following posterior probabilities; an asterisk (*) denotes 100% support, a hyphen (-) denotes less than 50% support. This is part of an analysis utilizing 384 sequences and represents all taxa belonging to the Leptolyngbyales sensu [22]. Strains from Yellow-stone National Park are in red font. The uncollapsed tree is available in the Supplemental Materials (Figure S1).

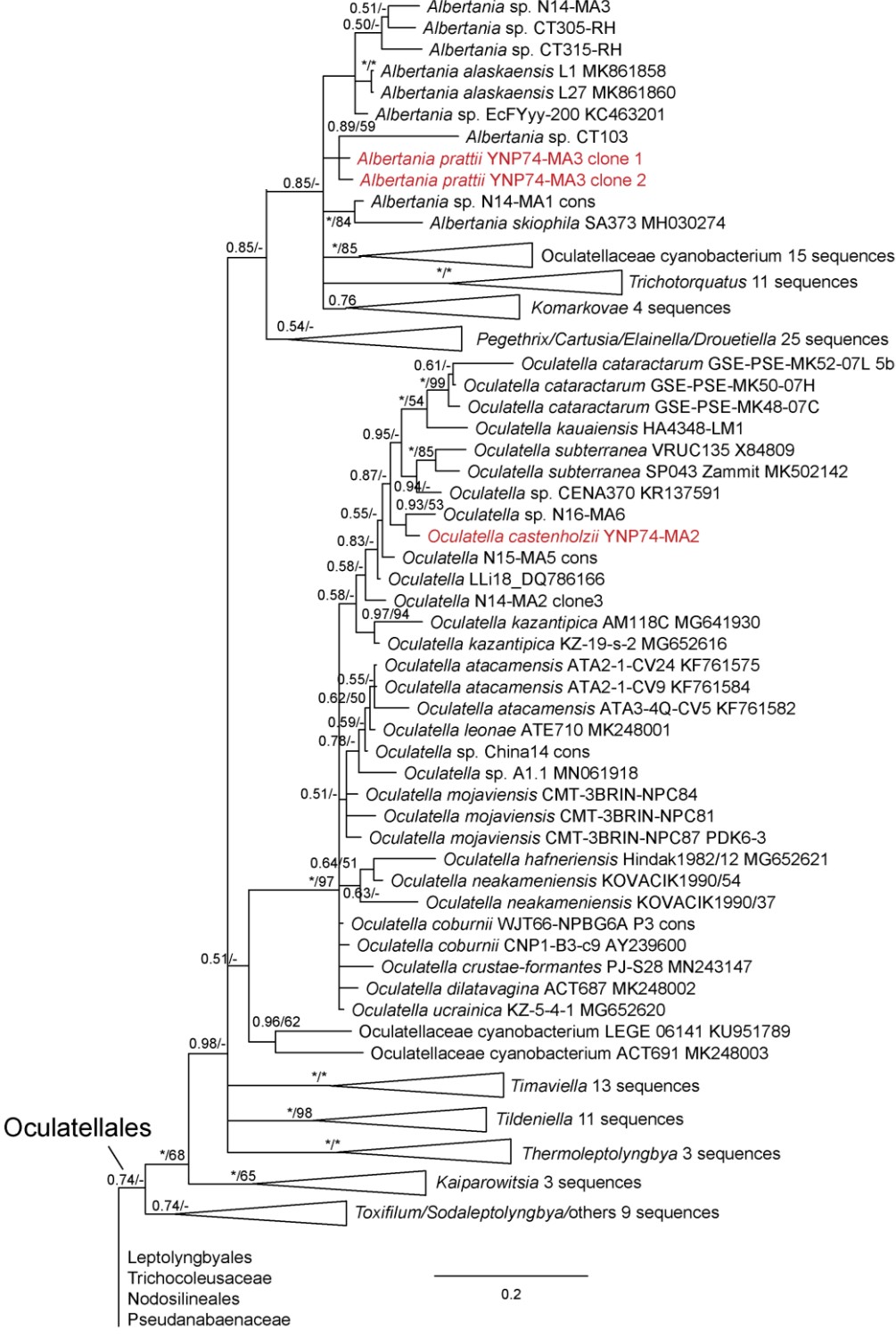

**Figure 3.** Bayesian inference analysis based on 16S rRNA gene sequence with maximum likelihood bootstrap values mapped to nodes following posterior probabilities; an asterisk (*) denotes 100% support, a hyphen (-) denotes less than 50% support. This is part of an analysis utilizing 384 sequences and represents all taxa belonging to the Oculatellales sensu [22]. Strains from Yellowstone National Park are in red font. The uncollapsed tree is available in Supplemental Materials (Figure S1).

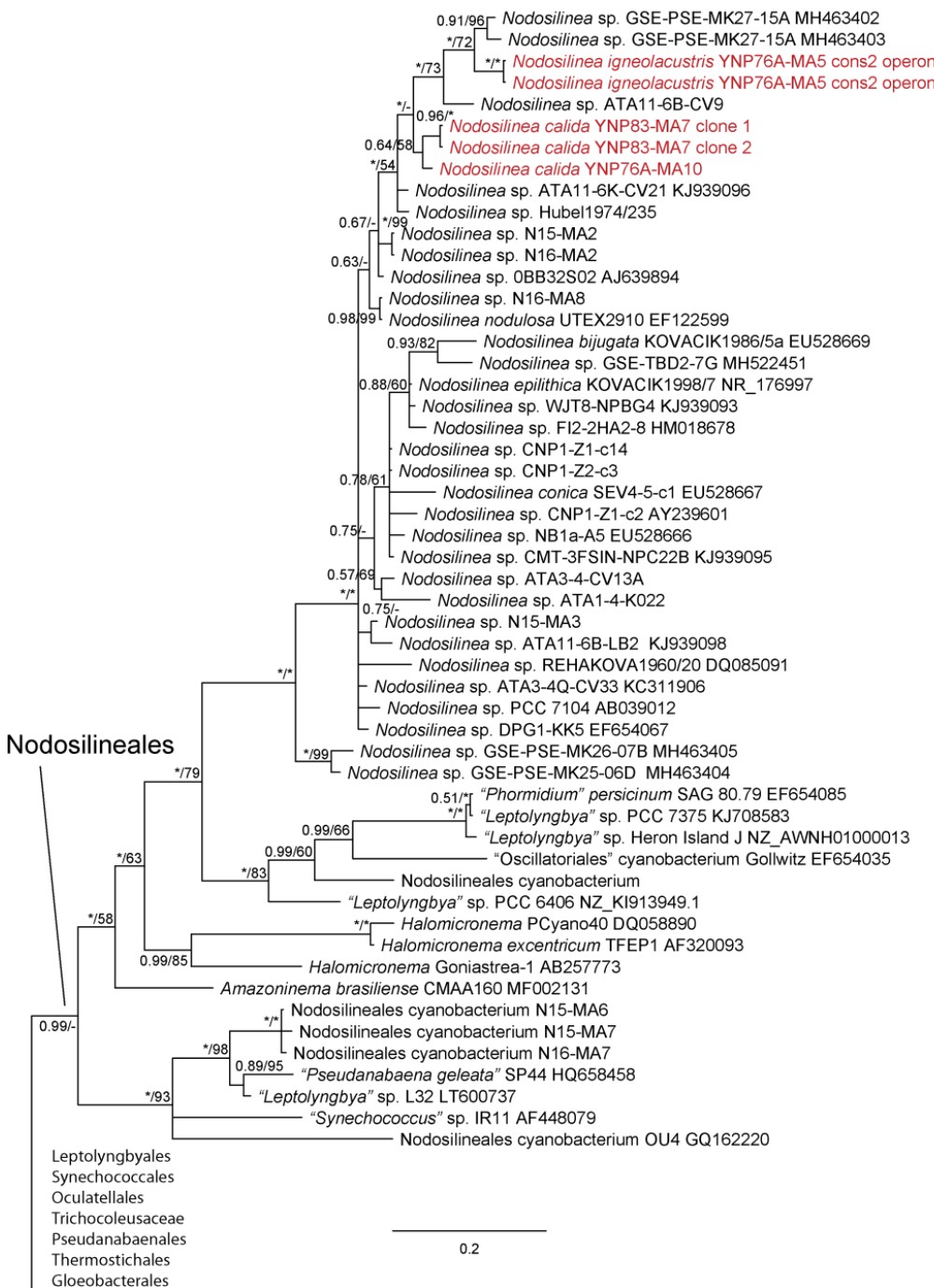

**Figure 4.** Bayesian inference analysis based on 16S rRNA gene sequence with maximum likelihood bootstrap values mapped to nodes following posterior probabilities; an asterisk (*) denotes 100% support, a hyphen (-) denotes less than 50% support. This is part of an analysis utilizing 384 sequences and represents all taxa belonging to the Nodosilineales sensu [22]. Strains from Yellowstone National Park are in red font. The uncollapsed tree is available in Supplemental Materials (Figure S1).

Taxonomic descriptions and diagnoses
Phylum: Cyanobacteria
Class: Cyanophyceae
Order: Leptolyngbyales
Family: Leptolyngbyaceae
***Copelandiella* Johansen, J.R., Kaštovský, J. & Akagha, M.U., gen. nov.**

Thallus formed of blue-green or green filaments, entangled in macroscopic mats. Filaments long, straight, or flexuous, without false branching. Sheaths thin, colorless. Trichomes not attenuated, constricted at cross walls, without necridic cells, motile and with granulation. Apical cells rounded, rarely conical-rounded. Cells mostly longer than wide.

ETYMOLOGY: This genus is named in honor of Joseph J. Copeland (1907–1990), who authored the first and only monograph of cyanobacteria from Yellowstone National Park, USA [2].

TYPE SPECIES: *Copelandiella yellowstonensis* Johansen, Kaštovský & Akagha

***Copelandiella yellowstonensis* Johansen, J.R., Kaštovský, J. & Akagha, M.U. spec. nov, Figure** 5**.**

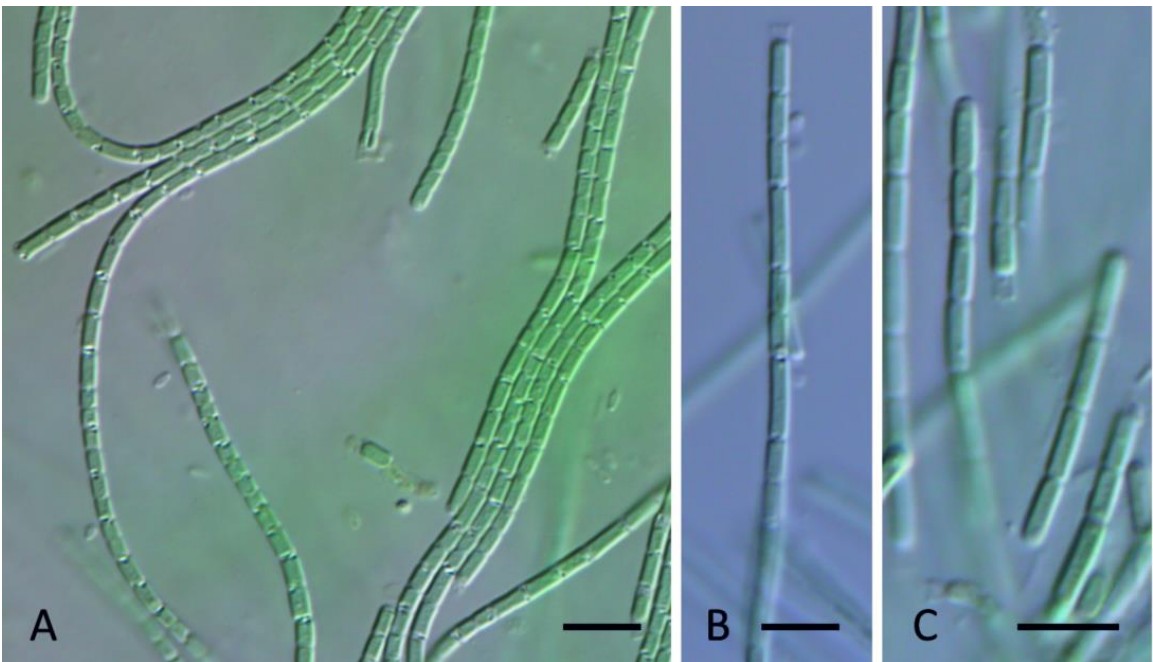

**Figure 5.** Light micrographs of *Copelandiella yellowstonensis* Johansen, J.R., Kaštovský, J. & Akagha, M. U. spec. nov. (**A**). Straight and flexuous filaments, with granulation on the cell walls; (**B**) Colorless, thin, sheaths; (**C**). Rounded apical cells. Scale bars = 5 μm.

DIAGNOSIS: Cryptic species, morphologically similar to other *Leptolyngbya*-like taxa. Distinguished by its isolated phylogenetic position in the Leptolyngbyaceae (Figure 2), its low 16S rRNA similarity to its closest sister taxa in other genera (mean = 93.2%, see Table 1), and its low 16S rRNA similarity to *C. thermalis* spec. nov. (<97.8 in all comparisons, Table 1). It is further separated from *C. thermalis* by its high dissimilarity to that taxon in 16S–23S ITS sequence (>20%, Table 2). ITS conserved domains are longer than in *C. thermalis*, with the D1-D1′ helix 12 nt longer, the Box-B helix 3 nt longer, and the V3 helix 6 nt longer (Figure 6). A variable V2 helix is present in *C. yellowstonensis* but absent in *C. thermalis* (Figure 6). The D1-D1′, Box-B, and V2 helices among populations of *C. yellowstonensis* were variable in sequence among populations of this species, but structural differences only occurred in the V2 helix. The V3 helix was absent in all cases.

**Table 1.** Percent similarity of 16S rRNA sequences among members of *Copelandiella* and its putative sister taxa. According to established microbial thresholds [40], percent similarity <98.7% is evidence for recognition of separate species, while <94.5% is evidence for separate genera.

| | YNP30-MA1 | YS71-RH2 | YNP81-MA3 | YNP81-MA4 | YNP83-MA6 | YN80B-MA3 | YN80B-MA4 | EY07 AM2 |
|---|---|---|---|---|---|---|---|---|
| *Copelandiella yellowstonensis* YNP30-MA1 | | | | | | | | |
| *Copelandiella yellowstonensis* YS71-RH2 | 99.8 | | | | | | | |
| *Copelandiella yellowstonensis* YNP81-MA3 | 99.7 | 99.8 | | | | | | |
| *Copelandiella yellowstonensis* YNP81-MA4 | 99.7 | 99.8 | 99.8 | | | | | |
| *Copelandiella yellowstonensis* YNP83-MA6 | 99.6 | 99.7 | 99.6 | 99.7 | | | | |
| *Copelandiella thermalis* YN80B-MA3 | 97.8 | 97.8 | 97.8 | 97.8 | 97.7 | | | |
| *Copelandiella thermalis* YN80B-MA4 | 97.8 | 97.8 | 97.8 | 97.8 | 97.7 | 100.0 | | |
| Leptolyngbyaceae cynobacterium EY07 AM2 | 91.5 | 91.5 | 91.5 | 91.5 | 91.4 | 90.9 | 90.9 | |
| Leptolyngbyaceae cynobacterium EY01 AM3 | 95.2 | 95.2 | 95.1 | 95.3 | 95.1 | 94.9 | 94.9 | 92.1 |

**Table 2.** Percent dissimilarity of 16S–23S ITS sequences among members of *Copelandiella* and its putative sister taxa. According to established microbial thresholds [41,42], percent dissimilarity >7% is strong evidence for recognition of separate species, while <3% is evidence for compared strains belonging to the same species. Values between 3 and 7% are ambiguous, and other evidence is required for recognition of separate species. See text for further explanation.

| | YNP81-MA1 | YNP81-MA4 | YNP70-MA1 | YNP83-MA6 | YNP30-MA1 | YNP80B-MA3 |
|---|---|---|---|---|---|---|
| *Copelandiella yellowstonensis* YNP81-MA1 | | | | | | |
| *Copelandiella yellowstonensis* YNP81-MA4 | 0.0 | | | | | |
| *Copelandiella yellowstonensis* YNP70-MA1 | 6.9 | 6.9 | | | | |
| *Copelandiella yellowstonensis* YNP83-MA6 | 7.6 | 7.6 | 1.1 | | | |
| *Copelandiella yellowstonensis* YNP30-MA1 | 7.5 | 7.5 | 4.4 | 3.6 | | |
| *Copelandiella thermalis* YNP80B-MA3 | 20.2 | 20.2 | 20.3 | 20.1 | 20.8 | |
| *Copelandiella thermalis* YNP80B-MA4 | 20.2 | 20.2 | 20.3 | 20.1 | 20.8 | 0.0 |

DESCRIPTION: Colony a mat, with entangled filaments, blue-green or green. Filaments long, straight, or flexuous, without false branching, 1.2–1.5 µm wide. Sheaths thin, colorless. Trichomes not narrowed toward the end, constricted at cross walls, without necridic cells, with motility, with granulation (conspicuous at the cross walls, Figure 5A), gray- or blue-green. Apical cells rounded, rarely conical rounded (Figure 5B,C). Cells mostly longer than wide, (1) 1.1–1.3 µm wide, (1.5) 2.0–3.2 (5.5) µm long.

HOLOTYPE: Dried material preserved in a permanently inactive state at Herbarium for Nonvascular Cryptogams at the Department of Botany, Faculty of Science, University of South Bohemia, Czech Republic), under the code CBFS A184-1!.

TYPE LOCALITY: Black Warrior Lake (Yellowstone National Park, WY, USA), 44.5444544 N, 110.7864528 W. Collected on 18 September 2019 by Jeffrey R. Johansen, Jan Kaštovský, and Jan Mareš.

HABITAT: Blackish crust under the boardwalk with rushing hot water, slightly submersed in water ~40 °C.

ETYMOLOGY: *yellowstonenis* = after the locality of the original sample.

REFERENCE STRAIN: YNP81-MA1, available as CCALA 10,292 (Culture Collection of Autotrophic Organisms at the Institute of Botany, Třeboň, CZ), isolated by Mildred U. Akagha.

GENE SEQUENCES: GenBank accession numbers OR259141-145 for the 16S rRNA and 16S–23S rRNA ITS genes.

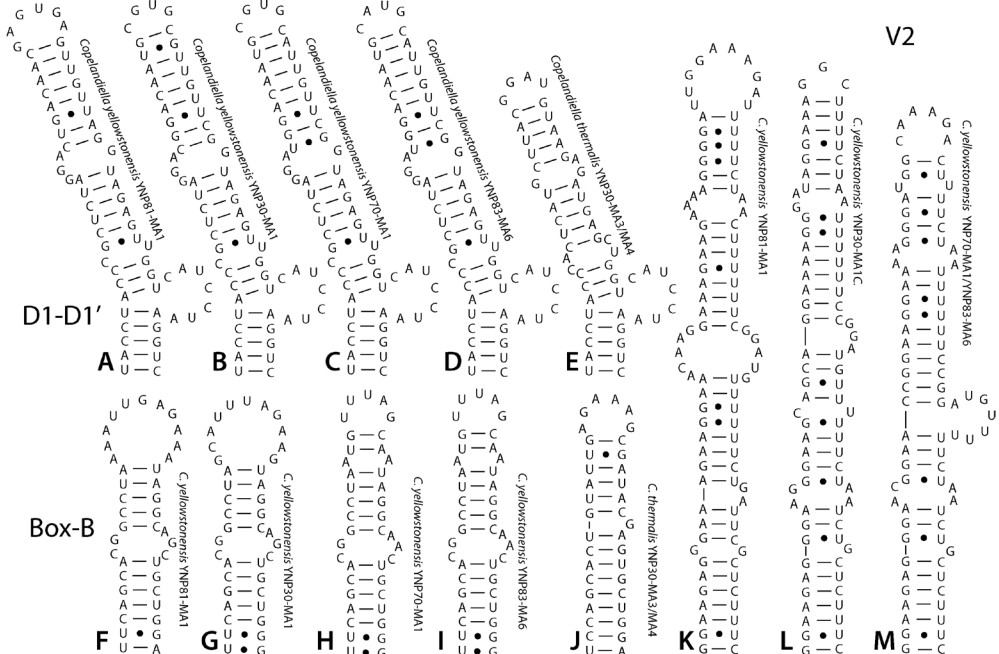

**Figure 6.** Secondary structure of conserved domains in the 16S–23S ITS of both *Copelandiella* species. (**A–E**). D1-D1′ helix. (**F–J**). Box-B helix. (**K–M**). V2 helix. *C. thermalis* lacked tRNA genes and therefore lacks a V2 helix. YNP81-MA1 is the reference strain for *C. yellowstonensis* (**A,F,K**). See text for explanation.

TAXONOMIC NOTES: In addition to the type locality, this species was also found in several other sites: a second site in Black Warrior Lake (YNP83), in hot water in Fairy Meadows (YNP30), and as an atmophyte close to Clearwater Springs (YNP70). These other isolates were grouped together based on high 16S rRNA similarity (>99.6% in all comparisons). ITS dissimilarity was indicative of greater diversity among these strains, with ITS percent dissimilarity being 6.9–7.6%, above the established threshold for distinguishing species. Strains from the type locality were notably different than strains from the other sites. YNP30 differed from YNP70 and YNP83 by 3.6–4.4%, which could indicate species separation for this population as well. However, given the high 16S rRNA similarity, and the morphological and ecological similarity, we cannot rule out the possibility that these differences in ITS sequence are due to multiple ribosomal operons, and we choose to include all strains in this species for the present, until further evidence of lineage separation can be obtained.

***Copelandiella thermalis* Kaštovský, J., Johansen, J.R. & Akagha, M.U. spec. nov., Figure 7.**

DIAGNOSIS: Morphologically similar to *C. yellowstonenis*. Distinguished by smaller granulation in cytoplasm (none visible at cross walls) and shorter cells. Clear separation was also indicated both in percent similarity of 16S rRNA gene sequence (Table 1) and percent dissimilarity among 16S–23S rRNA ITS sequence (Table 2). Furthermore, secondary structures of all conserved domains in the ITS distinctly differ (Figure 6).

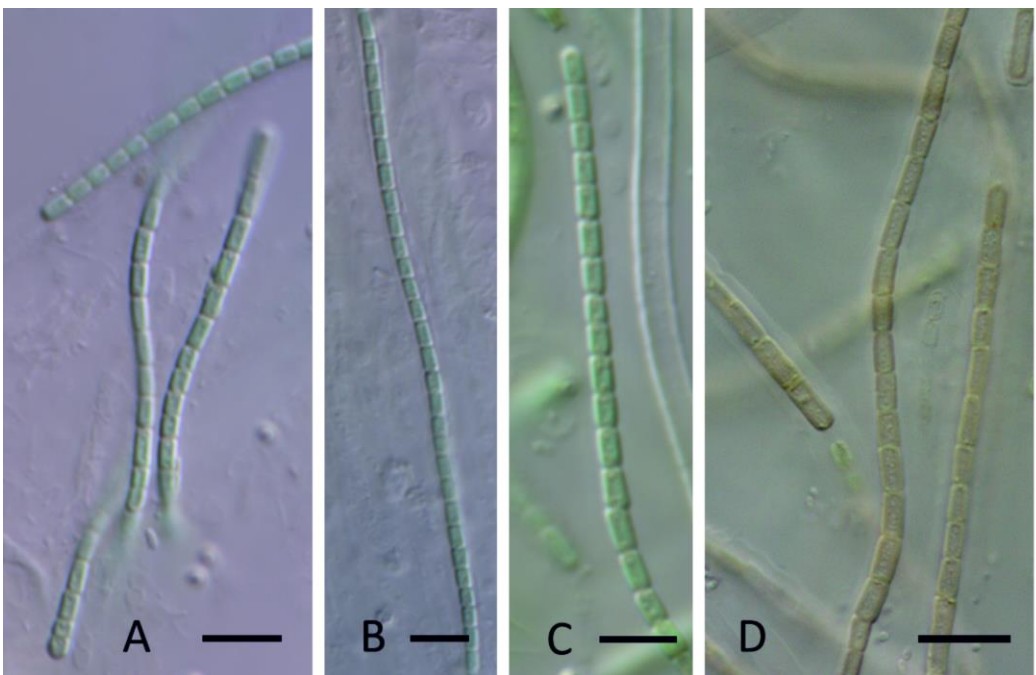

**Figure 7.** Light micrographs of *Copelandiella thermalis* Kaštovský, J., Johansen, J.R. & Akagha, M. U. spec. nov., (**A**). Filament straight or slightly flexuous, with clear constriction on cell walls; (**B**). Colorless, thin, sheaths; (**C**). Apical cells rounded; (**D**). Old filaments sometime brownish. Scale bars = 5 μm.

DESCRIPTION: Colony a mat, with entangled filaments, blue green or green. Filaments long, straight, or flexuous, without false branching, 1.3–1.5 μm wide (Figure 7A). Sheaths thin, colorless (Figure 7B). Trichomes not narrowed toward the end, constricted at cross walls, without necridic cells, with motility and with small granulation in cytoplasm, gray- or blue-green, rarely brownish-green (Figure 7C,D). Apical cells rounded, rarely conical-rounded. Cells mostly longer than wide, 1.2–1.3 μm wide, (1.5) 1.7–2.5 μm long.

HOLOTYPE: Dried material preserved in a permanently inactive state at Herbarium for Nonvascular Cryptogams at the Department of Botany, Faculty of Science, University of South Bohemia, Czech Republic), under the code CBFS A185-1!

TYPE LOCALITY: Black Warrior Lake (Yellowstone National Park, Wyoming, USA), 44.5444544 N, 110.7864528 W. Collected on 18 September 2019 by Jeffrey R. Johansen, Jan Kaštovský and Jan Mareš.

HABITAT: green mats, submersed at the bottom of hot lake.

ETYMOLOGY: *thermalis* = living in hot water.

REFERENCE STRAIN: YNP80B-MA3, isolate lost. Isolated by Mildred Akagha.

GENE SEQUENCES: GenBank accession numbers OR259139-140 for the 16S rRNA and 16S–23S rRNA ITS genes.

TAXONOMIC NOTES: Two strains were isolated from sample YNP80B, but unfortunately both were lost. Fortunately, morphological characterization, sequencing, and voucher preparation were completed before isolate failure.

*Kovacikia anagnostidisii* **Kaštovský, J., Johansen, J.R. & Hauerová, R. spec. nov., Figure 8.**

DIAGNOSIS: Morphologically differs from other *Kovacikia* species by constriction at cell walls and wider cells. Only *K. brockii* spec. nov. is wider (2.7 μm). Genetically distinguished by both 16S rRNA sequence (Table 3) and 16S–23S rRNA ITS sequence data (Table 4). Differing from *K. brockii* by having a V2 helix, shorter Box-B helix, and shorter V3 helix. Differing from *Leptothermofonsia sichuanensis* Daroch, Thang & Shah [43] in having D1-D1′ helix 58 nucleotides shorter. *K. muscicola* Miscoe, Pietrasiak & Johansen [35] and *K. minuta* Li-Qin Shen, Renhui Li & Qiu [44] are non-thermal.

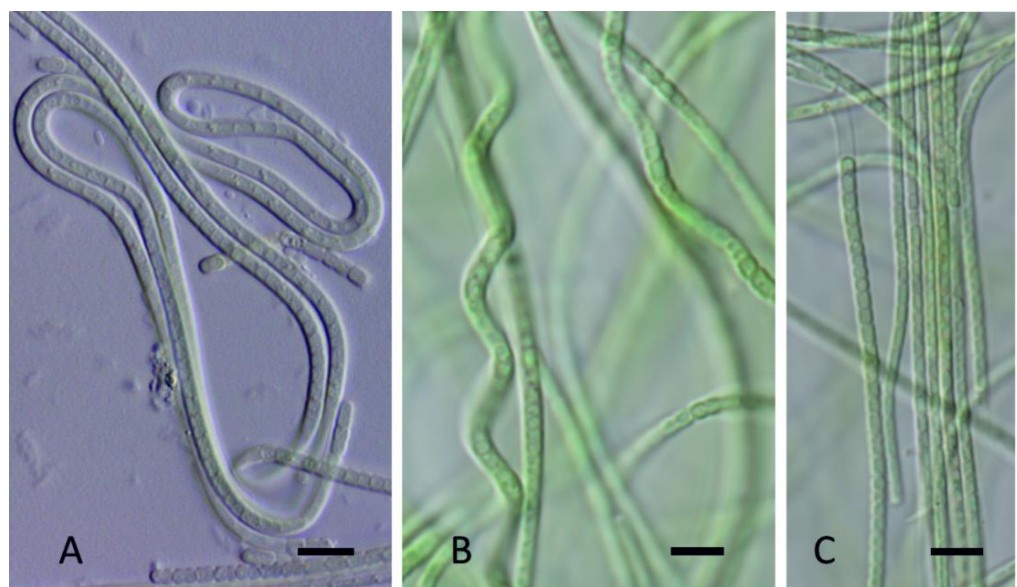

**Figure 8.** Light micrographs of *Kovacikia anagnostidisii* Kaštovský, J., Johansen, J.R. & Hauerová, R. spec. nov. (**A**). Filaments slightly constricted on the cell walls, coiled; (**B**). Filament spirally coiled or straight, with clear constriction on the cell walls; (**C**). Apical cells rounded, sheaths thin, colorless, attached to the trichome, almost invisible. Scale bars = 5 μm.

**Table 3.** Percent similarity of strains assigned to *Leptothermofonsia*, *Kovacikia*, unidentified or misidentified strains belonging to the clade and an unnamed sister taxon. Values < 94.5% generally indicate different genera, while values < 98.7% are typically considered different species. Members of the *Kovacikia* clade are indicated by light orange shading, darker orange shading indicates same species. New species described here appear in red font.

| | *Leptothermofonsia sichuanensis* | Leptolyngbyaceae CENA37 | Leptolyngbyaceae CENA37 | Leptolyngbyaceae CENA35 | *Leptolyngbya Greenland 10* | *Kovacikia brockii YS7-RH2* | *Kovacikia brockii YNP74-RH1* | *Kovacikia muscicola HA7619-LM3* | *Kovacikia anagnostidisii YS86-RH1* | *Kovacikia anagnostidisii YS89-RH1* | *Kovacikia minuta CCNU0001* | *Kovacikia atmophytica BACA0619* |
|---|---|---|---|---|---|---|---|---|---|---|---|---|
| *Leptothermofonsia sichuanensis* | | | | | | | | | | | | |
| Leptolyngbyaceae CENA37 | 95.4 | | | | | | | | | | | |
| Leptolyngbyaceae CENA37 | 95.5 | 99.9 | | | | | | | | | | |
| Leptolyngbyaceae CENA35 | 95.3 | 99.7 | 99.8 | | | | | | | | | |
| Leptolyngbya Greenland 10 | 96.4 | 96.6 | 96.7 | 96.7 | | | | | | | | |
| *Kovacikia brockii YS7-RH2* | 98.0 | 95.5 | 95.6 | 95.4 | 96.2 | | | | | | | |
| *Kovacikia brockii YNP74-RH1* | 97.9 | 95.5 | 95.6 | 95.4 | 96.2 | 99.9 | | | | | | |
| *Kovacikia muscicola* HA7619-LM3 | 95.2 | 98.8 | 98.9 | 98.7 | 95.8 | 95.4 | 95.4 | | | | | |
| *Kovacikia anagnostidisii YS86-RH1* | 96.1 | 96.8 | 96.9 | 96.7 | 94.8 | 96.0 | 96.0 | 97.3 | | | | |
| *Kovacikia anagnostidisii YS89-RH1* | 96.0 | 96.8 | 96.9 | 96.7 | 94.8 | 96.0 | 96.0 | 97.3 | 100 | | | |
| *Kovacikia minuta* CCNU0001 | 95.1 | 97.8 | 97.9 | 97.9 | 95.9 | 95.1 | 95.1 | 97.6 | 96.8 | 96.8 | | |
| *Kovacikia atmophytica* BACA0619 | 95.9 | 96.8 | 96.9 | 96.7 | 95.6 | 95.4 | 95.4 | 97.1 | 98.2 | 98.2 | 97.1 | |
| Leptolyngbyaceae MTQ1-MM2 | 94.3 | 94.3 | 94.4 | 94.3 | 93.7 | 94.6 | 94.6 | 94.8 | 96.1 | 96.1 | 94.0 | 95.3 |

**Table 4.** Percent dissimilarity among strains designated *Kovacikia* and *Leptothermofonsia* based upon the 16S–23S ITS sequence data. Values > 7.0% are considered strong evidence that compared strains are different species, whereas values < 3.0% are considered indicative of strains belonging to the same species. New species described here appear in red font.

| | *Leptothermofonsia sichuanensis* | *Kovacikia brockii*YNP74-RH1 | *Kovacikia brockii*YS7-RH2 | *Kovacikia muscicola* HA7619-LM3 | *Kovacikia anagnostidisii*YS86-RH1 | *Kovacikia anagnostidisii*YS89-RH1 | *Kovacikia atmophytica* BACA0619 |
|---|---|---|---|---|---|---|---|
| *Leptothermofonsia sichuanensis* | | | | | | | |
| *Kovacikia brockii*YNP74-RH1 | 18.8 | | | | | | |
| *Kovacikia brockii*YS7-RH2 | 18.8 | 0.0 | | | | | |
| *Kovacikia muscicola* HA7619-LM3 | 23.9 | 20.6 | 20.6 | | | | |
| *Kovacikia anagnostidisii*YS86-RH1 | 28.5 | 24.0 | 24.0 | 18.5 | | | |
| *Kovacikia anagnostidisii*YS89-RH1 | 28.5 | 24.0 | 24.0 | 18.5 | 0.0 | | |
| *Kovacikia atmophytica* BACA0619 | 26.5 | 21.2 | 21.2 | 12.0 | 18.2 | 18.2 | |
| *Kovacikia minuta* CCNUW1 | 26.1 | 21.2 | 21.2 | 17.0 | 24.1 | 24.1 | 17.3 |

DESCRIPTION: Colony a flat mat, with entangled filaments, green to yellow-green. Filaments slightly coiled, sometimes spirally coiled (Figure 8A–C), without false branching, 1.5–2.5 µm wide. Sheaths thin, colorless, attached to the trichome, scarcely visible. Trichomes not attenuated, distinctly constricted at cross walls, lacking necridia, immotile, granulated, grey-green or blue-green (Figure 8A–C). Apical cells rounded. Cells mostly shorter than wide or almost isodiametric, (1.3) 1.6–2.0 (2.4) µm wide and 1–2.1 (2.5) µm long.

HOLOTYPE: Dried material preserved in a permanently inactive state at Herbarium for Nonvascular Cryptogams at the Department of Botany, Faculty of Science, University of South Bohemia, Czech Republic), under the code CBFS A186-1!.

TYPE LOCALITY: Clearwater springs (Yellowstone National Park, WY, USA), 44.7886475 N, 110.7392136 W. Collected on 18 September 2019 by Jeffrey R. Johansen, Jan Kaštovský and Jan Mareš.

HABITAT: atmophytic, in massive ropy crust close to water ca. 85 °C.

ETYMOLOGY *anagnostidisii* = in honor of Konstantinos Th. Anagnostidis (1924–1994), Greek phycologist who studied thermal springs [45].

REFERENCE STRAIN: YS86-RH1, available as CCALA 10,299 (Culture Collection of Autotrophic Organisms at the Institute of Botany, Třeboň, CZ), isolated by Radka Hauerová.

GENE SEQUENCES: GenBank accession numbers OR236708-709 for the 16S rRNA and 16S–23S rRNA ITS genes.

TAXONOMIC NOTES: Two additional strains from Clearwater Springs were sequenced, YS89-RH1 and YNP90-MA1.

***Kovacikia brockii* Johansen, J.R., Kaštovský, J. & Hauerová, R. spec. nov., Figure 9.**

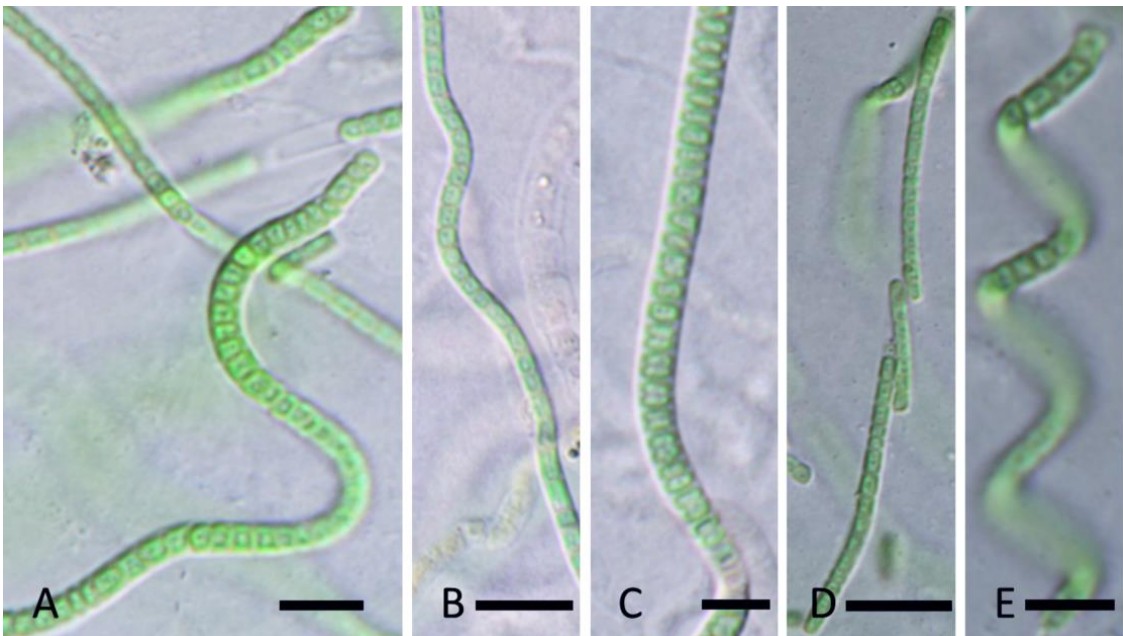

**Figure 9.** Light micrographs of *Kovacikia brockii* Johansen, J.R., Kaštovský, J. & Hauerová, R. spec. nov. (**A**). Filaments straight or flexuous; (**B**). Younger, thinner filament; (**C**). Older, wider filament; (**D**). Disintegrating into shorter trichomes, (hormogonia?); (**E**). Filaments are often spiral. Scale bars = 5 μm.

DIAGNOSIS: Morphologically differs from other *Kovacikia* species partly by the greater width of the cells (the other thermal species from Yellowstone, *K. anagnostidisii* spec. nov., is maximally 2.4 μm wide, while *K. muscicola*, *K. minuta*, and *Kovacikia* (*Leptothermofonsia*) *sichuanensis* are all less than 1.7 μm wide). Phylogenetically separated from other species (Figure 2). Genetically distinguished by both 16S rRNA (Table 3) and 16S–23S rRNA ITS (Table 4).

DESCRIPTION: Colony a flat mat, with entangled filaments, grey-green to blue-green. Filaments straight, coiled, sometimes spiral (Figure 9E), without false branching, 1.3–2.7 μm wide (Figure 9A,B). Sheaths thin, colorless, attached to the trichome (Figure 9A). Trichomes not narrowed toward the end, constricted or slightly constricted at cross walls (Figure 9B,C)., often disintegrating into hormogonia in the absence of necridia (Figure 9D), lacking motility, often with granulation in the middle of cell, pale blue-green to green. Apical cells rounded. Cells more or less isodiametric in younger filaments, or shorter in old trichomes, (0.9) 1.1–2.5 μm wide, 0.8–2.2 μm long, older trichomes are distinctly wider that young trichomes (Figure 9A).

HOLOTYPE: Dried material preserved in a permanently inactive state at Herbarium for Nonvascular Cryptogams at the Department of Botany, Faculty of Science, University of South Bohemia, Czech Republic), under the code CBFS A187-1!.

TYPE LOCALITY: Firehole Lake (Yellowstone National Park, WY, USA), 44.5440433 N, 110.7865600 W. Collected on 18 September 2019 by Jeffrey R. Johansen, Jan Kaštovský and Jan Mareš.

HABITAT: atmophytic mats near shore of the hot water.

ETYMOLOGY: *brockii* = named after the prominent microbial physiologist Thomas D. Brock, who wrote early works concerning microbial extremophiles in the hot springs of Yellowstone National Park [4].

REFERENCE STRAIN: YNP74-RH1, available as CCALA 10,294 (Culture Collection of Autotrophic Organisms at the Institute of Botany, Třeboň, CZ), isolated by Radka Hauerová.

GENE SEQUENCES: GenBank accession numbers OR220391-193, OR236707 for the 16S rRNA and 16S–23S rRNA ITS genes.

TAXONOMIC NOTES: In addition to the type locality, also found in Mound Spring as an atmophytic crust on the bank close to a stream (strain YS7-RH2).

*Leptolyngbya tildenae* **Kaštovský, J., Johansen, J.R. & Akagha, M.U. spec. nov., Figure** 10.

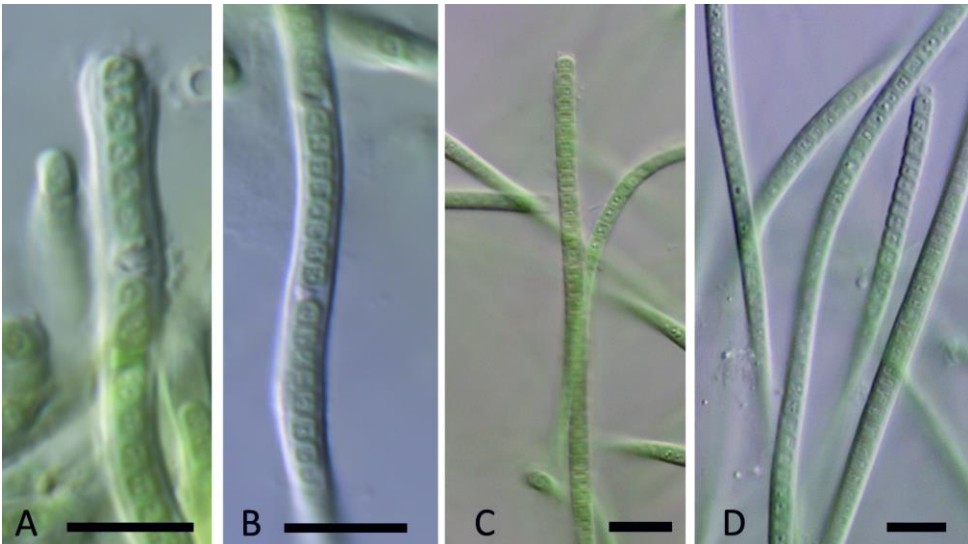

**Figure 10.** Light micrograph *Leptolyngbya tildenae* Kaštovský, J., Johansen, J.R. & Akagha, M. u. spec. nov. (**A**). Sheaths sometime widened at the end of filament, apical cells rounded; (**B**). Sheaths mostly attached to the trichome; (**C**). Filaments straight or slightly coiled, cells mostly shorter then wide; (**D**). Filaments slightly constricted or constricted at cross walls. Scale bars = 5 μm.

DIAGNOSIS: Morphologically similar to several nonthermal *Leptoplyngbya/Plectolyngbya* species, but the combination of morphological features with specific ecology is unique. It differs from its sister species *L. atmophytica* spec. nov. partly in size (it is thinner), but there is overlap. Genetically distinguished from *Tapinothrix clintonii clintonii* Bohunická & Johansen [46], the sister taxon to *Leptolyngbya* sensu stricto, by having percent similarity of the 16S rRNA gene <93.5% (Table 5). Differentiated from all described *Leptolyngbya* and *Plectolyngbya* species by having 16S–23S rRNA ITS sequence dissimilarity >9.0. (Table 6). Phylogenetic separation from other species was also seen (Figure 11), with the sister species to *L. tildenae* being *L. vaporiphila* sp. nov. and an unnamed species from soils of Joshua Tree National Park (WJT66-NPBG17).

DESCRIPTION: Colony a flat mat, with entangled filaments, dark green. Filaments straight or slightly coiled (Figure 10C), without false branching, 1.6–2.2 μm wide. Sheaths firm, thick, colorless, attached to the trichome or widened at the end (Figure 10A,B). Trichomes not narrowed toward the end, constricted or unconstricted at cross walls (Figure 10D), necridic cells not observed, without motility, with granulation, blue-green. Apical cells rounded. Cells mostly shorter than wide or almost isodiametric, 1.4–1.9 (2.2) μm wide, 1.3–1.8 (2) μm long.

HOLOTYPE: Dried material preserved in a permanently inactive state at Herbarium for Nonvascular Cryptogams at the Department of Botany, Faculty of Science, University of South Bohemia, Czech Republic), under the code CBFS A188-1!.

TYPE LOCALITY: Firehole Lake, (Yellowstone National Park, WY, USA), 44.5440433 N, 110.7865600 W. Collected on 18 September 2019 by Jeffrey R. Johansen, Jan Kaštovský and Jan Mareš.

HABITAT: Black mat from soil on the bank of hot water lake.

ETYMOLOGY: *tildenae* = in honor of Josephine Elisabeth Tilden (1869–1957), American phycologist, who worked on the Yellowstone algal flora [47].

REFERENCE STRAIN: YNP77-MA3, available as CCALA 10,300 (Culture Collection of Autotrophic Organisms at the Institute of Botany, Třeboň, CZ), isolated by Mildred U. Akagha.

**Table 5.** Percent similarity of 16S rRNA gene sequences among *Albertania* species. All strains have 16S rRNA sequences <98.7%, indicating presence of different species. *A. prattii* had two operons that differed slightly in the 16S rRNA gene sequence.

| | *A. prattii* YNP74-MA3.1 | *A. prattii* YNP74-MA3.2 | *A. skiophila* SA373 | *A. alaskaensis* L1 | *Albertania* N14-MA1 | *Albertania* N14-MA3 | *Albertania* EcFYyy-200 | *Albertania* CT103 | *Albertania* CT115 | *Albertania* CT305 |
|---|---|---|---|---|---|---|---|---|---|---|
| *A. prattii* YNP74-MA3.1 | | | | | | | | | | |
| *A. prattii* YNP74-MA3.2 | 99.4 | | | | | | | | | |
| *A. skiophila* SA373 | 96.6 | 96.6 | | | | | | | | |
| *A. alaskaensis* L1 | 98.1 | 98.0 | 96.6 | | | | | | | |
| *Albertania* N14-MA1 | 98.0 | 97.9 | 98.1 | 97.9 | | | | | | |
| *Albertania* N14-MA3 | 96.6 | 96.5 | 96.9 | 97.8 | 97.7 | | | | | |
| *Albertania* EcFYyy-200 | 97.5 | 97.4 | 97.2 | 98.8 | 97.9 | 98.2 | | | | |
| *Albertania* CT103 | 97.8 | 97.7 | 96.9 | 97.1 | 97.7 | 96.1 | 97.0 | | | |
| *Albertania* CT115 | 95.1 | 95.0 | 93.6 | 95.2 | 95.2 | 94.8 | 94.8 | 95.0 | | |
| *Albertania* CT305 | 96.5 | 96.4 | 97.0 | 97.8 | 97.4 | 98.6 | 98.4 | 96.6 | 94.7 | |
| *Albertania* CT315 | 96.2 | 96.1 | 96.1 | 97.0 | 96.6 | 97.6 | 97.5 | 95.9 | 95.8 | 97.5 |

**Table 6.** Percent dissimilarity of ITS regions among *Albertania* species. All strains have pairwise comparisons with other strains that indicate they belong to separate species (PD >7.0). However, note two distinct clonal sequences were found for *A. prattii* YNP74-MA3, which almost surely indicate the presence of two operons.

| | *A. prattii* YNP74-MA3.1 | *A. prattii* YNP74-MA3.2 | *A. skiophila* SA373 | *A. alaskaensis* KV23 | *Albertania* sp. N14-MA1 | *Albertania* sp. N14-MA3 | *Albertania* sp. CT103 | *Albertania* sp. CT305 | *Albertania* sp. CT315 |
|---|---|---|---|---|---|---|---|---|---|
| *Albertania prattii* YNP74-MA3.1 | | | | | | | | | |
| *Albertania prattii* YNP74-MA3.2 | 16.7 | | | | | | | | |
| *Albertania skiophila* SA373 | 13.1 | 22.5 | | | | | | | |
| *Albertania alaskaensis* KV23 | 15.2 | 18.7 | 19.4 | | | | | | |
| *Albertania* sp. N14-MA1 | 13.7 | 16.3 | 15.0 | 15.3 | | | | | |
| *Albertania* sp. N14-MA3 | 14.0 | 16.7 | 12.5 | 17.5 | 8.0 | | | | |
| *Albertania* sp. CT103 | 14.7 | 16.1 | 13.8 | 17.3 | 7.8 | 8.1 | | | |
| *Albertania* sp. CT305 | 11.5 | 17.2 | 10.6 | 13.5 | 9.8 | 12.0 | 9.6 | | |
| *Albertania* sp. CT315 | 14.3 | 19.6 | 15.3 | 20.8 | 14.5 | 11.8 | 12.9 | 11.2 | |
| *Albertania* sp. CT115 | 11.2 | 26.3 | 20.6 | 14.8 | 15.8 | 13.2 | 13.8 | 16.2 | 23.1 |

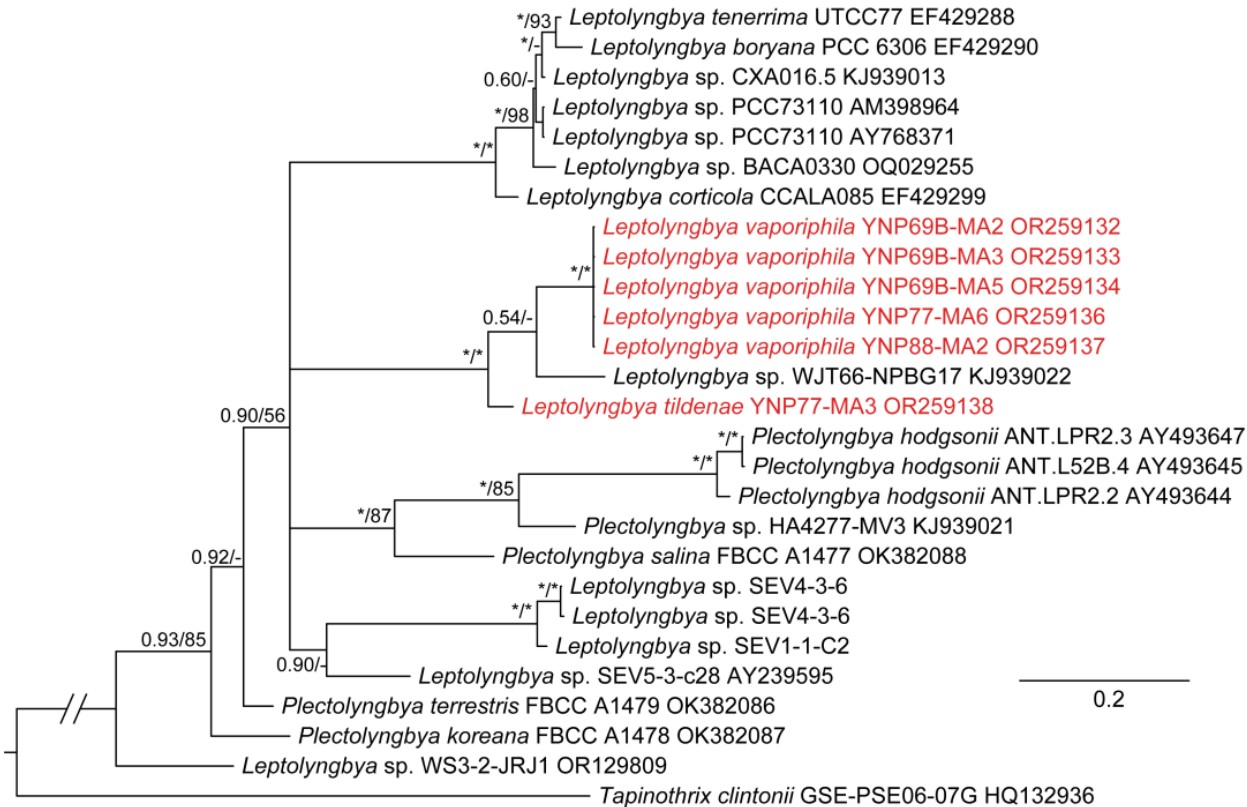

**Figure 11.** Bayesian inference analysis based on aligned 16S–23S ITS sequences with maximum parsimony bootstrap values mapped to nodes following posterior probabilities; an asterisk (*) denotes 100% support, a hyphen (-) denotes less than 50% support. This cluster of strains contains the type species of *Leptolyngbya*, *L. boryana*, and consequently all members of the clade belong to *Leptolyngbya* sensu stricto. See text for further explanation.

GENE SEQUENCES: GenBank accession number OR259138 for the 16S rRNA and 16S–23S rRNA ITS genes.

***Leptolyngbya vaporiphila*** **Johansen, J.R., Kaštovský, J., & Akagha, M.U. spec. nov., Figure 12.**

DIAGNOSIS: Morphologically similar to several nonthermal *Leptolyngbya* / *Plectolyngbya* species, but the combination of morphological features with specific ecology is unique. It differs from its sister species *L. tildenae* spec. nov. partly in size (it is wider), although there is some overlap. Differentiated from all sequenced *Leptolyngbya* and *Plectolyngbya* species by having 16S–23S rRNA ITS sequence dissimilarity >9.0. (Table 6). Phylogenetic separation from other species was also evident (Figure 11).

DESCRIPTION: Colony a flat mat, with entangled filaments, dark green. Filaments straight or slightly coiled (Figure 12A,B), with rare false branching (Figure 12C), 2–3.2 μm wide. Sheaths firm, thick, colorless, attached to the trichome (Figure 12A). Trichomes not attenuated, constricted or unconstricted at cross walls, with necridia (Figure 12A,B), lacking motility, with granulation, blue-green. Apical cells rounded. Cells mostly shorter than wide or almost isodiametric, (1.7) 2.0–3.0 μm wide, 1.6–2.1 μm long.

HOLOTYPE: Dried material preserved in a permanently inactive state at Herbarium for Nonvascular Cryptogams at the Department of Botany, Faculty of Science, University of South Bohemia, Czech Republic), under the code CBFS A189-1!.

TYPE LOCALITY: Clearwater springs (Yellowstone National Park, WY, USA), 44.7886475 N, 110.7392136 W. Collected on 18 September 2019 by Jeffrey R. Johansen, Jan Kaštovský and Jan Mareš.

HABITAT: atmophytic, in massive ropy crust close to water ca 85 °C.

ETYMOLOGY: *vaporiphila* = living with constant presence of hot vapor.

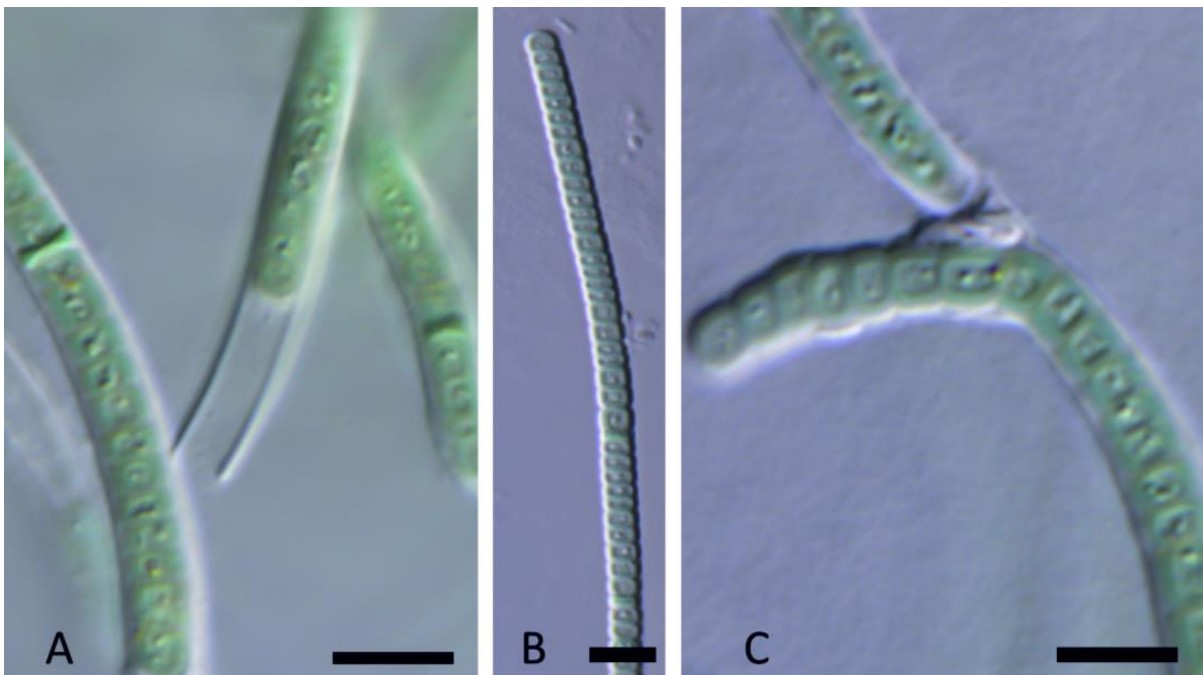

**Figure 12.** Light micrograph *Leptolyngbya vaporiphila* Johansen, J.R., Kaštovský, J., & Akagha, M. spec. nov. (**A**). Filament without constriction on the cell walls and with necridia and firm, thick, and colorless sheaths; (**B**). Filaments straight or slightly coiled, with constriction at the cell walls, apical cells rounded; (**C**). False branching, cells with granulation. Scale bar = 5 μm.

REFERENCE STRAIN: YNP69B-MA2, available as CCALA 10,295 (Culture Collection of Autotrophic Organisms at the Institute of Botany, Třeboň, CZ), isolated by Mildred Akagha.

GENE SEQUENCES: GenBank accession numbers OR259132-137 for the 16S rRNA and 16S–23S rRNA ITS genes.

TAXONOMIC NOTES: In addition to the type locality, this species was also found as an atmophyte in several places on the bank of Firehole Lake.

Phylum: Cyanobacteria
Class: Cyanophyceae
Order: Oculatellales
Family: Oculatellaceae

***Albertania pratii* Johansen, J.R., Kaštovský, J. & Akagha, M.U. spec. nov., Figure 13.**

DIAGNOSIS: Sole thermal species in genus *Albertania*. Morphologically similar to both described *Albertania* species, cell width is larger than *A. alaskaensis* Strunecký, Raabová, Bernardová, Ivanova, Semanova, Crossley & Kaftan [48] and slightly smaller than *A. skiophila* Zammit [37]. In contrast to *A. skiophila*, trichomes of *A. prattii* have no necridic cells. Distinguished by its thermal ecology. Pairwise comparisons of 16S rRNA gene sequence with other species and strains are all <98.7% (Table 5). The percent dissimilarity of the 16S–23S ITS region was high among all strains, >11% (Table 6).

DESCRIPTION: Colony a flat mat, with entangled filaments, blue-green. Filaments normally straight or slightly waived, without false branching, 1.9–2.3 μm wide. Sheaths thin, colorless, attached to the trichome, often widened at the end (Figure 13A,B). Trichomes not narrowed toward the end, slightly constricted at cross walls, without necridia, lacking motility, with granulation, pale green. End cells rounded, sometimes with more evident constriction at cross walls, lacking granulation (Figure 13C–E). Hormogonia few celled, motile (Figure 13E). Cells mostly isodiametric or slightly longer than wide (1.3) 1.7–2.1 μm wide, (1.7) 2.3–3.2 μm long.

HOLOTYPE: Dried material preserved in a permanently inactive state at Herbarium for Nonvascular Cryptogams at the Department of Botany, Faculty of Science, University of South Bohemia, Czech Republic), under the code CBFS A190-1!

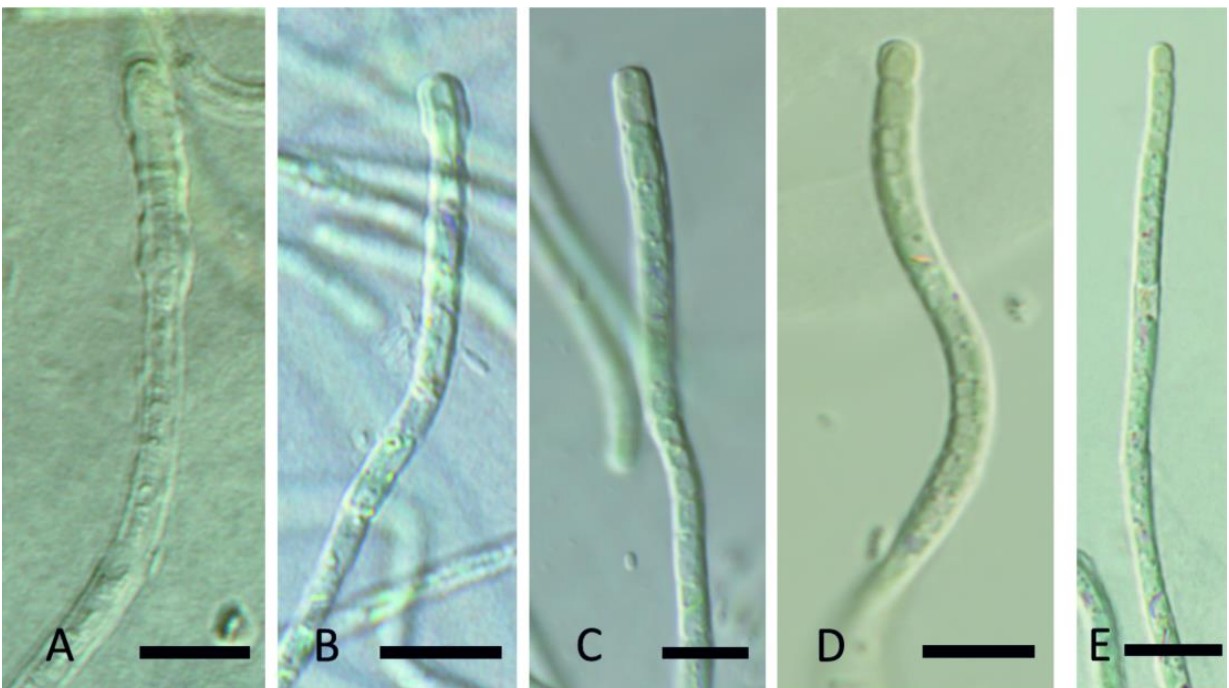

**Figure 13.** Light micrograph *Albertania prattii* Johansen, J.R., Kaštovský, J. & Akagha, M. spec. nov. (**A**,**B**). Filaments straight or slightly waived, sheath often widened at the end; (**C**,**D**). Apical cells rounded, sometimes with clearer constriction at the cross walls and without granulation; (**E**). Hormogonia few celled, sheath thin, colorless. Scale bars = 5 μm.

TYPE LOCALITY: Firehole Lake, (Yellowstone National Park, WY, USA), 44.5440433 N, 110.7865600 W. Collected on 18 September 2019 by Jeffrey R. Johansen, Jan Kaštovský, and Jan Mareš.

HABITAT: atmophytic near shore, not submersed, brown velvet mat.

ETYMOLOGY: *pratii* = in honor of Silvestr Prát (1895–1990), Czech phycologist, who worked on the Yellowstone algal flora [49].

REFERENCE STRAIN: YNP74-MA3, available as CCALA 10,298 (Culture Collection of Autotrophic Organisms at the Institute of Botany, Třeboň, CZ), isolated by Mildred U. Akagha.

GENE SEQUENCES: GenBank accession numbers OR244603, OR259146 for the 16S rRNA and 16S–23S rRNA ITS genes.

***Oculatella castenholzii* Kaštovský, J., Johansen, J.R. & Akagha, M.U. spec. nov., Figure 14.**

DIAGNOSIS: The only known described thermal species of the whole genus *Oculatella*. Phylogenetically most similar to several tropical unnamed strains from Nigeria (N16-MA6, N15-MA5, N14-MA2) and Costa Rica (LLi8) (Figure 3). Phylogenetically closest to *O. hafneriensis* according to 16S–23S rRNA ITS sequence analysis (Figure 15). Below the 16S rRNA similarity threshold for separation of species (98.7%) for all strains and species except *Oculatella* sp. LLi8 collected from a thermal spring in Costa Rica, which has a sequence similarity of 99.1%. It is plausible that the Costa Rican strain and our strain are conspecific, but we do not have 16S–23S ITS data for this strain, so this conclusion is not presently supported, as we cannot include it in our phylogenetic analysis (Figure 15) or our pairwise comparisons of the 16S–23S ITS dissimilarity. The pairwise comparisons with all strains for

which ITS sequence data exist were >5.2%, indicating *O. castenholzii* is a separate species from all presently described species.

DESCRIPTION: Colony a flat mat, with entangled filaments, blue green. Filaments long, straight, or flexuous (Figure 14A,B), without false branching, 1.0–1.2 µm wide. Sheaths thin, colorless, attached to the trichome. Trichomes not narrowed toward the end, slightly constricted at cross walls (Figure 14C,D), without necridia, without motility, without granulation, pale blue-green. Apical cells conical rounded, often with orange carotenoid inclusion at the tip (Figure 14A,B,D). Cells longer than wide (0.8) 0.9–1.1 µm wide, 2.0–4.2 (6) µm long.

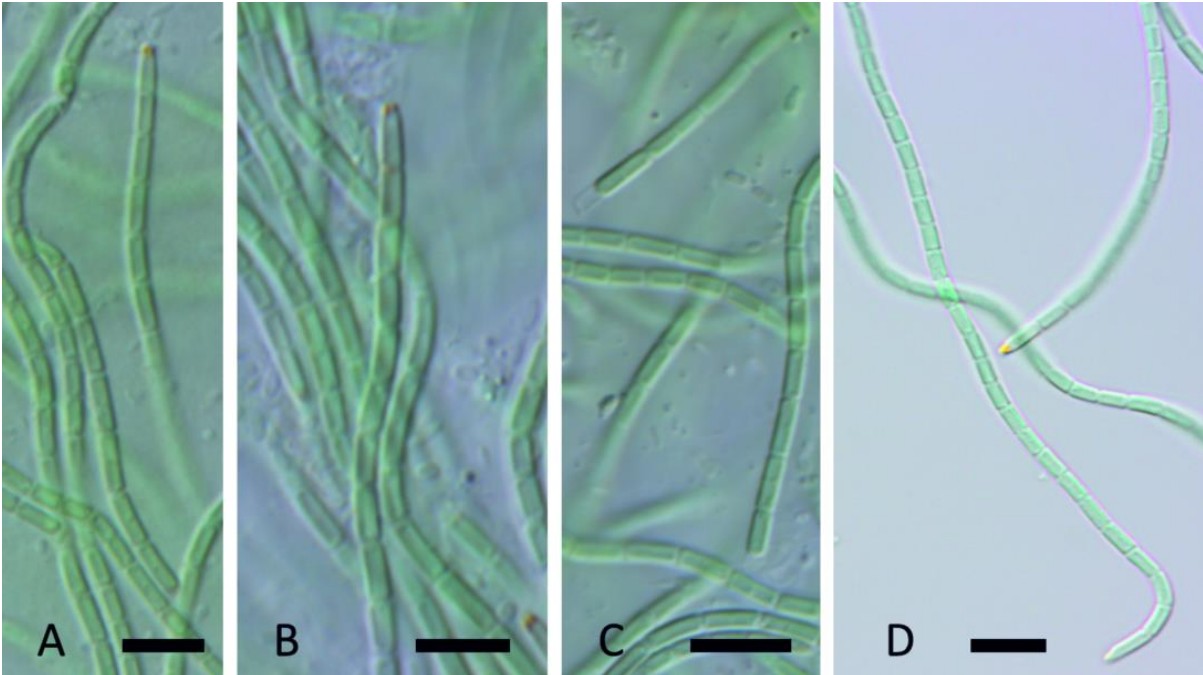

**Figure 14.** Light micrograph *Oculatella castenholzii* Kaštovský, J., Johansen, J.R. & Akagha, M. spec. nov. (**A**,**B**). Filaments straight or flexuous often with orange carotenoid inclusion at the tip, cells with constriction on the cell walls; (**C**). Sheaths thin, colorless, attached to the trichome; (**D**). Apical cells conical rounded. Scale bars = 5 µm.

HOLOTYPE: Dried material preserved in a permanently inactive state at Herbarium for Nonvascular Cryptogams at the Department of Botany, Faculty of Science, University of South Bohemia, Czech Republic), under the code CBFS A191-1!.

TYPE LOCALITY: Firehole Lake, (Yellowstone National Park, WY, USA), 44.5440433 N, 110.7865600 W. Collected on 18 September 2019 by Jeffrey R. Johansen, Jan Kaštovský, and Jan Mareš.

HABITAT: atmophytic near shore, not submersed, part of brown velvet mat.

ETYMOLOGY: *castenholzii* = in honor of Richard W. Casteholz (1931–2018), American phycologist, who worked on Yellowstone algal flora.

REFERENCE STRAIN: YNP74-MA2, available as CCALA 10,293 (Culture Collection of Autotrophic Organisms at the Institute of Botany, Třeboň, CZ), isolated by Mildred Akagha.

GENE SEQUENCES: GenBank accession number OR259147 for the 16S rRNA and 16S–23S rRNA ITS genes.

Phylum: Cyanobacteria

Class: Cyanophyceae

Order: Nodosilineales

Family: Nodosilineaceae

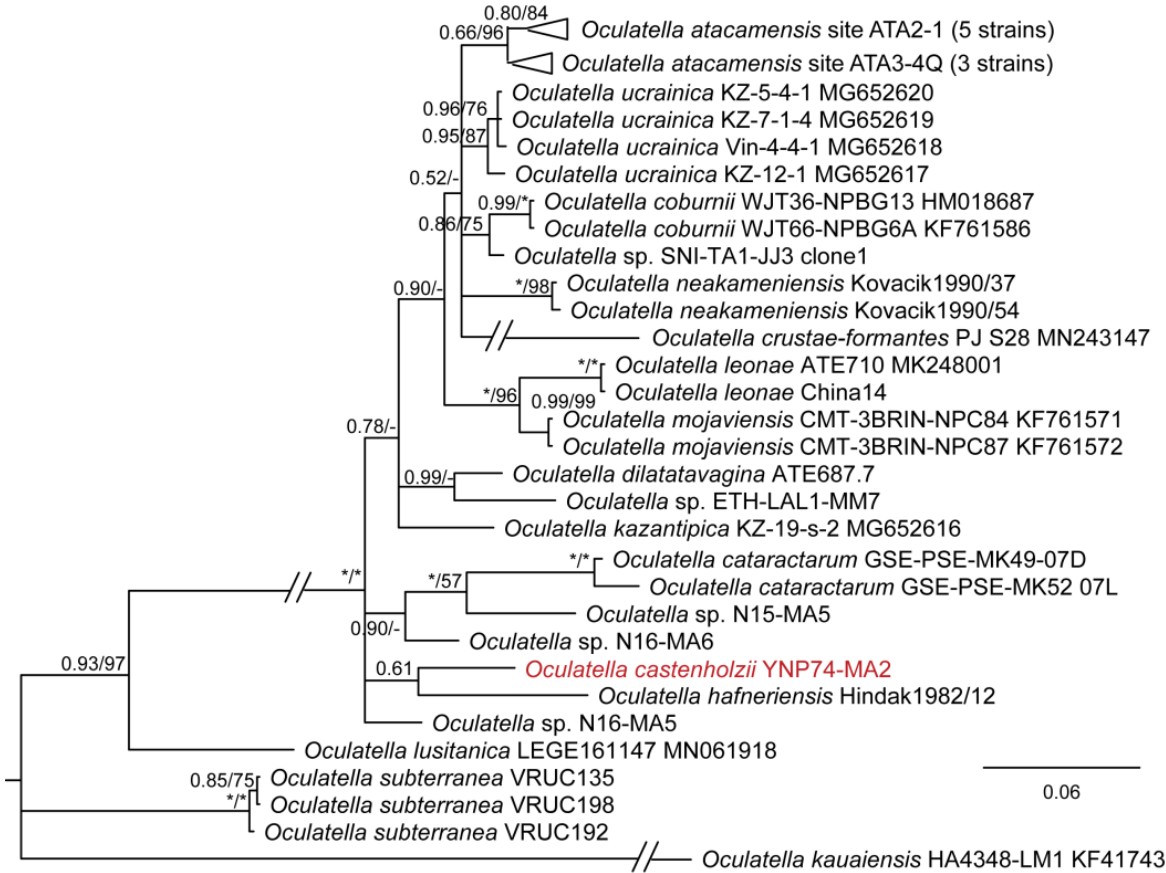

**Figure 15.** Bayesian inference analysis based on aligned 16S–23S ITS sequences with maximum parsimony bootstrap values mapped to nodes following posterior probabilities; an asterisk (*) denotes 100% support, a hyphen (-) denotes less than 50% support. This cluster of strains contains the type species of *Oculatella*, *O. subterranea*, and consequently all members of the clade belong to *Oculatella* sensu stricto. See text for further explanation.

***Nodosilinea calida* Johansen, J.R., Kaštovský, J. & Akagha, M.. spec. nov., Figure 16.**
　　DIAGNOSIS: One of only two thermal species in the whole genus *Nodosilinea*. *N. igneolacustris* spec. nov. and *N. calida* spec. nov. are sister taxa, differing only slightly by length of cells (*N. igneolacustris* has a maximum length of cells 2.5 μm, *N. calida* can reach 3, exceptionally up to 4, μm). These two thermal species are distinct in both the 16S rRNA phylogeny and the 16S–23S ITS phylogeny. Furthermore, they are <98.7% similar in the ITS region, and >15% dissimilar in their aligned 16S–23S ITS sequences, so by both threshold criteria they meet the standard for being recognized as separate species. In comparison to other strains in the genus, they are closest to GSE-PSE-MK27-15A, ATA11-6B-CV9, and ATA1106K-CV21 (Figure 4), but these are all unnamed and at the 16S rRNA similarity threshold for species (98.6–99.1% similar). They exceed the ITS dissimilarity threshold (>12% in comparisons with these taxa). *N. calida* is separated from *N. igneolacustris* according to the 16S–23S ITS phylogeny (Figure 17), and both taxa have long branches. We lack ITS data for some of the close taxa identified in the 16S rRNA phylogeny (Figure 4).

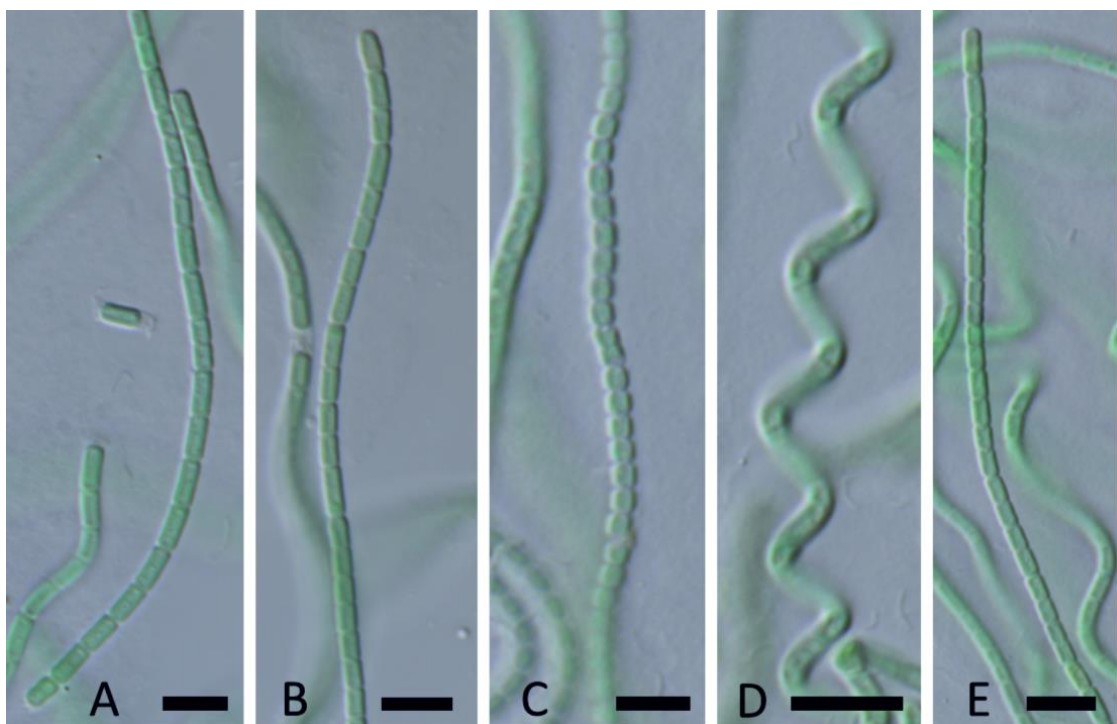

**Figure 16.** Light micrograph *Nodosilinea calida* Johansen, J.R., Kaštovský, J. & Akagha, M. spec. nov. (**A**). Filaments long, straight, or flexuous; (**B**). Sheaths thin, colorless, attached to the trichome, often almost invisible. (**C**). Cells slightly constricted at the cell walls, older filaments more constricted; (**D**). Filaments sometime spiral coiled; (**E**). Apical cells rounded, cells isodiametric or longer than wide. Scale bars = 5 μm.

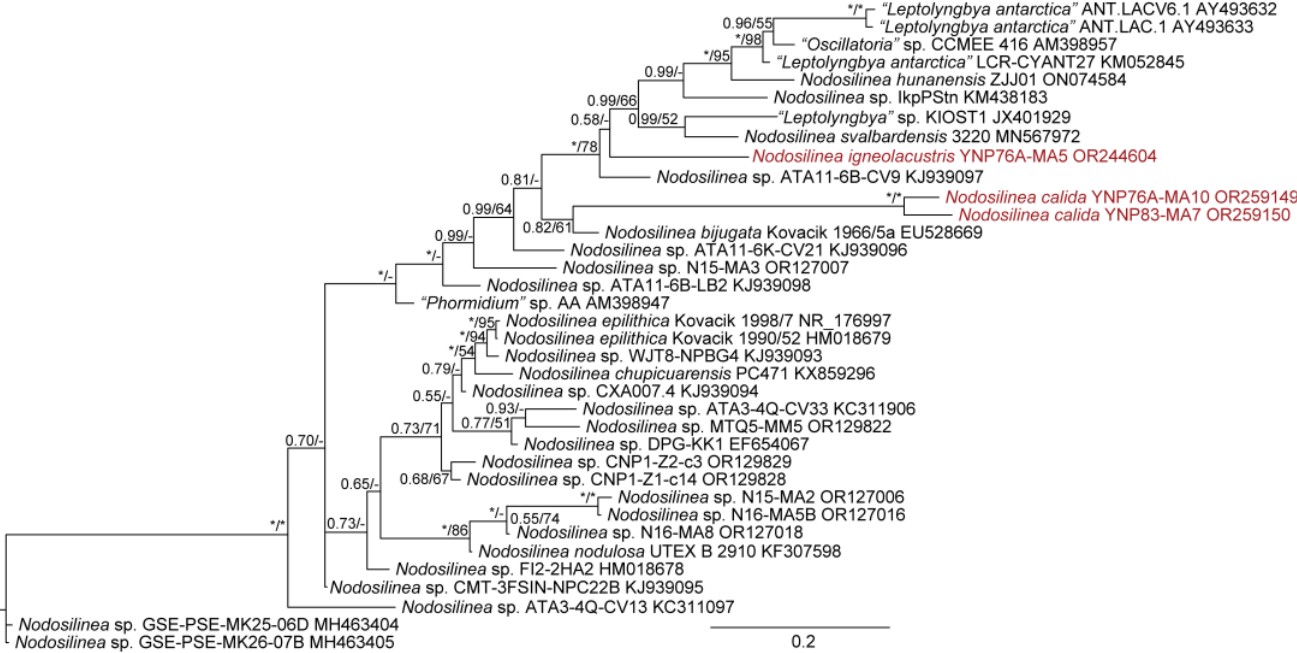

**Figure 17.** Bayesian inference analysis based on aligned 16S–23S ITS sequences with maximum parsimony bootstrap values mapped to nodes following posterior probabilities; an asterisk (*) denotes 100% support, a hyphen (-) denotes less than 50% support. This cluster of strains contains the type species of *Nodosilina*, *N. nodulosa*, and consequently all members of the clade belong to *Nodosilinea* sensu stricto. See text for further explanation.

DESCRIPTION: Colony a flat mat, with entangled filaments, dark green. Filaments long, straight, or flexuous, sometimes spiral (Figure 16A,D,E) without false branching, 1.3–1.5 μm wide. Sheaths thin, colorless, attached to the trichome, often almost invisible (Figure 16B). Trichomes not narrowed toward the end, constricted at cross walls (Figure 16B,C), without necridia, lacking motility, without granulation, blue-green. End cells rounded. Cells longer than wide or isodiametric (Figure 16B,C), 1.2–1.4 μm wide, 1.7–3.0 (4) μm long.

HOLOTYPE: Dried material preserved in a permanently inactive state at Herbarium for Nonvascular Cryptogams at the Department of Botany, Faculty of Science, University of South Bohemia, Czech Republic), under the code CBFS AA192-1!.

TYPE LOCALITY: Firehole Lake, (Yellowstone National Park, WY, USA), 44.5440433 N, 110.7865600 W. Collected on 18 September 2019 by Jeffrey R. Johansen, Jan Kaštovský, and Jan Mareš.

HABITAT: submersed or atmophytic calcareous concretions in pool with hot water by the road.

ETYMOLOGY: *calida* = from a hot environment.

REFERENCE STRAIN: YNP76AMA10, available as CCALA 10,296 (Culture Collection of Autotrophic Organisms at the Institute of Botany, Třeboň, CZ), isolated by Mildred Akagha.

GENE SEQUENCES: GenBank accession numbers OR259149-150 for the 16S rRNA and 16S–23S rRNA ITS genes.

TAXONOMIC NOTES: This species was found repeatedly in different locations within the type locality, which is quite large and ecologically variable, including a submerged population. Nodule formation, a diacritical character of the genus, was not observed.

***Nodosilinea igneolacustris* Kaštovský, J., Johansen, J.R. & Akagha, M.U, spec. nov., Figure 18.**

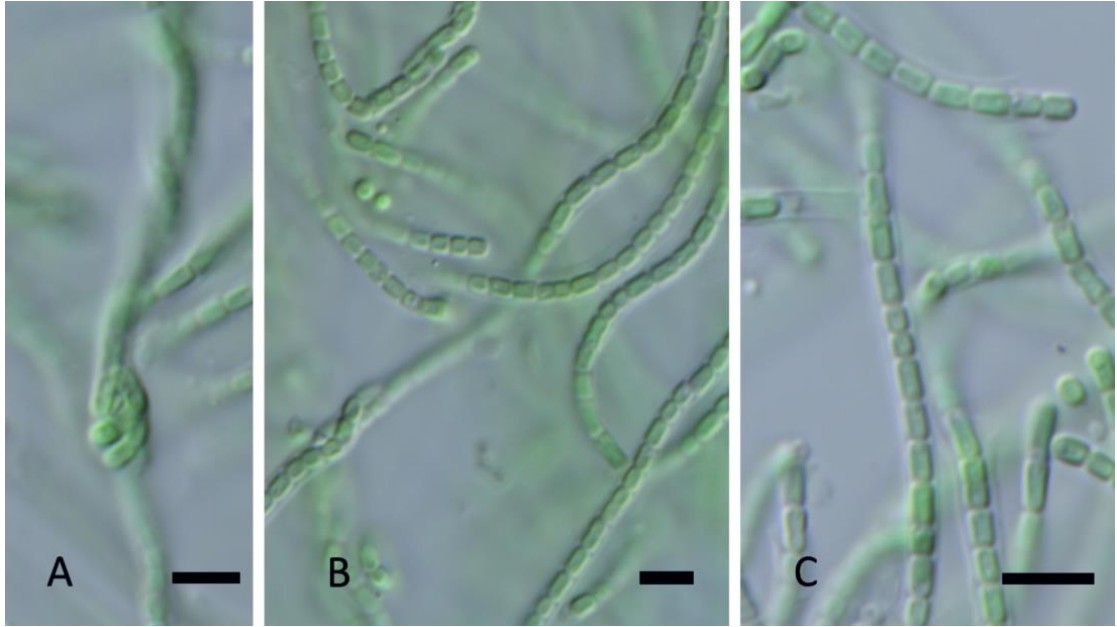

**Figure 18.** Light micrograph *Nodosilinea igneolacustris* Kaštovský, J., Johansen, J.R. & Akagha, M. spec. nov. (**A**). Nodules on filament; (**B**). Filaments straight or flexuous, trichomes constricted at cross wall, cells longer than wide or isodiametric; (**C**). Sheaths thin, colorless, attached to the trichome, apical cells rounded. Scale bars = 5 μm.

DIAGNOSIS: Thermal species genetically separated from all other described species (see above).

DESCRIPTION: Colony a flat mat, with entangled filaments, dark green. Filaments long, straight or flexuous, without false branching, with nodules (Figure 18A), 1.2–1.5 μm

wide. Sheaths thin, colorless, attached to the trichome, sometimes widened (Figure 18B,C). Trichomes not narrowed toward the end, constricted at cross walls, without necridia, lacking motility, without granulation, blue-green. End cells rounded (Figure 18C). Cells longer than wide or isodiametric, 1.1–1.4 μm wide, 1.5–2.5 μm long.

HOLOTYPE: Dried material preserved in a permanently inactive state at Herbarium for Nonvascular Cryptogams at the Department of Botany, Faculty of Science, University of South Bohemia, Czech Republic), under the code CBFS A193-1!.

TYPE LOCALITY: Firehole Lake, (Yellowstone National Park, WY, USA), 44.5440433 N, 110.7865600 W. Collected on 18 September 2019 by Jeffrey R. Johansen, Jan Kaštovský, and Jan Mareš.

HABITAT: submersed calcareous concretions in pool with hot water by the road.

ETYMOLOGY: *igneolacustris* = of the fire lake, referring to its occurrence in Firehole Lake.

REFERENCE STRAIN: YNP76AMA5, available as CCALA 10,297 (Culture Collection of Autotrophic Organisms at the Institute of Botany, Třeboň, CZ), isolated by Mildred Akagha.

GENE SEQUENCES: GenBank accession numbers OR244604, OR259148 for the 16S rRNA and 16S–23S rRNA ITS genes.

## 4. Discussion

Merging of genera

Although the formation of new genera has been much more frequent in recent decades than their merging, merging is not an isolated phenomenon [50]. Most often, this comes about because a subsequent larger set of molecular data show that the incomplete data available when the new taxon was postulated led to a conclusion that the larger dataset does not support, e.g., the collapsing of *Sphaerocavum* [51] or *Cylindrospermopsis* [52].

The same reasons motivated us to merge the two pairs of genera: *Leptothermofonsia* Daroch, Tang & Shah 2022 to *Kovacikia* Miscoe, Pietrasiak & Johansen 2016 and *Plectolyngbya* Taton, Wilmotte, Smarda, Elster & Komárek 2011 [53] to *Leptolyngbya* Anagnostidis & Komárek 1988.

In addition to respecting the priority rule in both cases (the older name should persist)—according to the International Code of Nomenclature for Algae, Fungi, and Plants [54]—the situation in the case of *Leptothermofonsia* was made even more complicated by the fact that it is not a valid name, due to improper designation of the holotype [23].

*Leptolyngbya* has long been a problematic genus, having been used as a catch-all for any thin filamentous taxon with a sheath. Over a hundred species were recognized in [23], and its polyphyletic character in numerous phylogenies has required extensive revision. All of the taxa covered in this paper would have been placed in *Leptolyngbya* 20 years ago. The genus should be narrowed to just those species belonging to the same clade as the type species, *L. boryana* (Gomont) Anagnostidis & Komárek [36]. This species has been sequenced and is quite close to other species in the genus, including *L. angustata* Casamatta & Johansen [55], *L. corticola Johansen*, Kováčik, Cassamata, Fučíková & Kaštovský [56], *L. foveolarum* (Gomont) Anagnostidis & Komárek [36] and *L. tenerrima* (Hansgirg) Komárek [57]. *Plectolyngbya* is phylogenetically very close to *Leptolyngbya*, but was initially recognized for its double false branching. However, *Leptolyngbya sensu stricto* has single false branching and similar cell structure (mostly isodiametric to shorter than wide). The two genera are above 94.5% similar in the majority of possible comparisons (Table S1). Furthermore, the sister taxon to the clade containing both *Plectolyngbya* and *Leptolyngbya* is *Tapinothrix clintonii*, and there is a marked discontinuity in 16S similarity between it and all members of the *Plectolyngbya/Leptolyngbya* cluster. Finally, many of the more recently studied *Plectolyngbya* do not have the diagnostic feature of double false branching. We conclude that *Plectolyngbya* is a later synonym of *Leptolyngbya*, and below will give an emended description of *Leptolyngbya* and complete the new combinations, so that the species described in that genus will now reside in *Leptolyngbya*. The clade that is created by uniting these taxa is *Leptolyngbya sensu stricto*. All other sequences occurring outside that clade but designated as *Leptolyngbya* require revision.

*Leptolyngbya* Anagnostidis & Komárek 1988.

SYNONYM: *Plectolyngbya* A.Taton, A.Wilmotte, J.Smarda, J.Elster & Komárek
EMENDED DESCRIPTION:

Filaments long, solitary or coiled into clusters and fine mats (which are sometimes macroscopic and several cm in diameter), arcuated, waved or intensely coiled, isopolar, thin, fine, 1.5–3.2 μm wide, with simple, thin, usually colorless facultative sheaths opened at the apical end; sheaths joined to the trichomes or slightly distant from them, enveloping only one, very rarely (in short sections) two trichomes; often with false branching. Trichomes fine, cylindrical, usually not attenuated to the ends or slightly attenuated, with rounded or conical apical cells, not constricted or constricted at the cross walls, immotile when encased in sheath. Cells isodiametrical, longer or shorter than wide, cylindrical, with homogeneous content, without aerotopes, rarely with scarce prominent granules, pale blue-green, greyish, olive-green, yellowish or reddish; end cells without thickened cell walls or calyptra. Heterocytes and akinetes absent. Cells divide by symmetrical (rarely asymmetrical) crosswise binary fission, cells grow to original size before next division. Reproduction by motile hormogonia, liberated at trichome ends, fragmented with or without necridic cells.

ECOLOGY: in soils, periphyton and metaphyton of freshwater and halophilous (marine) biotopes, thermal and mineral springs, aerophytic, endogloeic in mucilage of other algae.

TYPE SPECIES: *Leptolyngbya boryana* (Gomont) Anagnostidis & Komárek 1988: 391

New combinations:

*Leptolyngbya hodgsonii* (Taton, Wilmotte, Šmarda, Elster & Komárek) Johansen et Kaštovský, comb. nov.

BASIONYM: *Plectolyngbya hodgsonii* Taton, Wilmotte, Šmarda, Elster & Komárek 2011: 184, Figures 3–8 [53].

*Leptolyngbya koreana* (D.-H.Kim, N.-J.Lee, E.-C.Yang & O.-M.Lee) Johansen et Kaštovský, comb. nov.

BASIONYM: *Plectolyngbya koreana* D.-H.Kim, N.-J.Lee, E.-C.Yang & O.-M.Lee 2022: 7, Figures 4 and 5 [58].

*Leptolyngbya salina* (D.-H.Kim, N.-J.Lee, E.-C.Yang & O.-M.Lee) Johansen et Kaštovský, comb. nov.

BASIONYM: *Plectolyngbya salina* D.-H.Kim, N.-J.Lee, E.-C.Yang & O.-M.Lee 2022: 9, Figures 6 and 7.

*Leptolyngbya terrestris* (D.-H.Kim, N.-J.Lee, E.-C.Yang & O.-M.Lee) Johansen et Kaštovský, comb. nov.

BASIONYM: *Plectolyngbya terrestris* D.-H.Kim, N.-J.Lee, E.-C.Yang & O.-M.Lee 2022: 5, Figures 2 and 3.

Taxonomically more complicated is the situation in the genus *Leptothermofonsia*. This genus has not been validly described [23]—lacking type material in a physiologically inactive state, [45], nor can the species be transferred to *Kovacikia*, which is the name with priority. However, our results clearly show that this genus is not molecularly distinct from the genus *Kovacikia*. It is in a clade with several *Kovacikia* species, and it does not meet the similarity thresholds accepted by microbiologists. The occurrence of more thermal species in *Kovacikia* indicates that this criterion is not sufficient for recognizing a genus separate from *Kovacikia*. Furthermore, *Leptothermofonsia* is not morphologically distinct from *Kovacikia*. There would be no significant changes in the original description of the genus *Kovacikia* even with the inclusion of the species *Leptothermofonsia sichuanensis* Daroch, J.Tang & M.R.Shah, nom. inval.; perhaps only the inclusion of the possibility of trichomes unconstricted at the cross walls. If a nomenclaturally valid type were prepared and deposited in a curated collection, then the species epithet *sichuanensis* could be used in combination with the genus *Kovacikia*. We hope that the curators of these well-characterized strains will validate the species in the near future.

Morphological evaluation with existing species

Although many of the cyanobacterial species mentioned by Copeland were found during our sampling trip, this was not the case with these simple filaments. For the

most part, the morphology of the species we described does not match that described (or reported) by Copeland [2].

The species described within the genus *Phormidium* Kützing ex Gomont [59] could be considered for identity, the other filamentous taxa differ too much in the width of the filament (e.g., *Oscillatoria* Vaucher ex Gomont [59]) or the greater number of trichomes in the filament (e.g., *Schizothrix* Kützing ex Gomont [60]).

Many of the species formerly belonging to the genus *Phormidium* from localities in Yellowstone National Park have a pointed end cell (*Leptolyngbya laminosa* (Gomont) Anagnostidis & Komárek [36], *L. vesiculosa* (Copeland) Anagnostidis [57], *L. subcapitata* (J.B.Petersen) Anagnostidis [57] and *L. subuliformis* (Gomont) Anagnostidis [57], or even a gradually tapering end as in *L. fragilis* (Gomont) Anagnostidis & Komárek [36], *L. granulifera* (Copeland) Anagnostidis [57] or *L. tenuis* (Gomont) Anagnostidis & Komárek [36], which none of our species have.

The newly described species also differ from *L. cartilagina* (Copeland) Anagnostidis [57], because none of our species have granules on both sides of the cross walls. Other taxa also differ in color—ours are all blue- or grey-green and *Drouetiella lurida* (Gomont) Mai, Johansen & Pietrasiak [61] is brownish-violet. *Leptolyngbya rubra* (Tilden) Anagnostidis [57] is scarlet or pink. *L. purpurascens* (Gomont) Anagnostidis & Komárek [36] is violet.

*L. copelandii* Anagnostidis [27] (a synonym of *Phormidum truncatum* var. *thermale* Copeland [2]) and *L. ramosa* (J.B.Petersen) Anagnostidis & Komárek [36] are unconstricted at cross walls (our species are minimally at least sometimes constricted). *L. geysericola* (Copeland) Anagnostidis [57], *Phormidium bajahense* Copeland [2] and *Phormidium subterraneum* Phillipson [62] are too small (both with cell width max. 0.6 µm) and create pale salmon or yellowish expanded mats.

The only taxon from Copeland's list that could be morphologically somewhat similar to some of our species is *Leptolyngbya foveolarum* (Gomont) Anagnostidis & Komárek [36]—1.5 µm width of the cells, 0.8–2.0 µm length, not tapering, rounded apical cell, no granulation, pale blue-green trichomes. However, this description is so general that it can be identified with several different genera in today's understanding, and its phylogenetic position (within the *L. boryana* clade) is different from our strains. Furthermore, its habitat of origin is very different from thermal mineral waters.

Other cyanobacteria of similar morphological appearance are collected in the monograph by [63]. However, even here we do not find identical thermal species. Contrary to our species *Leptolyngbya gelatinosa* (Woronichin) Anagnostidis & Komárek [36] and *L. thermobia* Anagnostidis [57] have cells shorter than wide, *L. orientalis* (G.S.West) Anagnostidis & Komárek [36] is not constricted and possesses very long cells (to 8 µm). *L. thermarum* (Woronichin) Anagnostidis & Komárek [36] is not constricted and possesses solitary granules at cross walls. *L. phormidioides* (Anagnostidis) Anagnostidis & Komárek [36] has apical cells conically narrowed into a pointed apex and *L. treleasei* (Gomont) Anagnostidis & Komárek [36] is too small (0.4–0.8 µm wide).

We therefore proposed our strains as legitimate new species for science.

The work remaining

This manuscript describes 10 new species of simple filamentous cyanobacteria from Yellowstone National Park. This could certainly be just a beginning. Copeland [2] reported 81 new species from the hot springs of Yellowstone. He did his work without the benefit of molecular data. We only made isolation efforts in a few of the samples we collected and are certain there would be many more if a more systematic and longer-lasting effort was made. Much of the past work on Yellowstone cyanobacteria was ecophysiological in nature, and a modern taxonomic effort is needed. We had difficulty with some of our cultures, which failed before full characterization. However, based on our sequencing efforts, we know there were five more simple filamentous taxa that could have been described. A unique species from *Elainella* cluster was isolated twice, but in both cases became contaminated with another cyanobacterium. Two undescribed genera, one in

the *Scytolyngbya/Limnolyngbya* cluster in the Leptolyngbyales and one in the Oculatellales cluster sister to *Trichotorquatus* Pietrasiak & Johansen [64]. Finally, we have two new *Stenomitos* Miscoe & Johansen [35] species, which will be described in a separate treatment of that genus. This work demonstrates that there is ample opportunity for discovery of microbial diversity in extreme habitats such as those found in Yellowstone National Park.

**Supplementary Materials:** The following supporting information can be downloaded at: https://www.mdpi.com/article/10.3390/d15090975/s1, Table S1: Percent similarity of 16S rRNA sequences for the *Leptolyngbya/Plectolyngbya* clade; Table S2: Percent dissimilarity for aligned ITS regions in the *Leptolyngbya/Plectolyngbya* clade; Table S3: Percent dissimilarity for aligned ITS regions in the *Nodosilinea* clade; Figure S1: Uncollapsed BI tree based on aligned 16S rRNA gene sequences of 372 simple trichal cyanobacteria, with BI posterior probabilities and ML bootstrap support values mapped to the nodes.

**Author Contributions:** Conceptualization, methodology, writing, supervision, and project administration J.K. and J.R.J., sampling J.R.J. and J.K., material isolation and cultivation M.U.A. and R.H., morphological data J.K., molecular data M.U.A., R.H. and phylogenetic evaluation J.R.J. All authors have read and agreed to the published version of the manuscript.

**Funding:** This research was supported by grants from the Czech Ministry of Education project Inter Excellence LTAUSA 18008 and Grant Agency of the Czech Republic GAČR 22–06374S.

**Institutional Review Board Statement:** Not applicable.

**Data Availability Statement:** All taxa with Genbank accession numbers can be found at https://doi.org/10.6084/m9.figshare.23735901.v1 (accessed on 24 July 2023). The uncollapsed 16S rRNA phylogeny can found in Supplementary Materials, Figure S1, which also contains information about accession numbers.

**Acknowledgments:** The authors give thanks to Jan Mareš for help with sample collection, Alžběta Vondrášková for help with culture cultivation, and Jan Pokorný for help with obtaining molecular data.

**Conflicts of Interest:** The authors declare no conflict of interest.

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
