# Peer review of "Hot Is Rich—An Enormous Diversity of Simple Trichal Cyanobacteria from Yellowstone Hot Springs"

_diversity, doi:10.3390/d15090975_

Round 1

Reviewer 1 Report

This is a modern and nicely written paper, which describes new cyanobacterial taxa using a comprehensive approach. But I have several recommendations which should be solved.

You recognized the position of Leptothermofonsia as the member of Kovacikia genus and I agree that that clade is monophyletic, but could you add some sequences to your trees to estimate a position of following strains KR137595, KR137599, KR137576, KY924323, DQ431005?

Also, in my opinion, you able to consider the lineage of Leptolyngbya crispata AY239599, AY239598 and your described Leptolyngbya species.

Why didn't you use recently Nodosilinea taxa in your analysis? I mean, new species cited in

Davydov D., Shalygin S., Vilnet A. New cyanobacterium Nodosilinea svalbardensis sp. nov. (Prochlorotrichaceae, Synechococcales) isolated from alluvium in Mimer river valley of the Svalbard archipelago // Phytotaxa, 2020. Vol. 442(2). P. 61-79. DOI: https://doi.org/10.11646/phytotaxa.442.2.2

Cai, F.; Li, S.; Zhang, H.; Yu, G.; Li, R. Nodosilinea hunanesis sp. nov. (Prochlorotrichaceae, Synechococcales) from a Freshwater Pond in China Based on a Polyphasic Approach. Diversity 2022, 14, 364. https:// doi.org/10.3390/d14050364

Perhaps, your topology will change. Please, add sequence similarity comparison of the 16S rRNA gene between all the Nodosilinea species.

In figures 2-4 you indicated consensus sequences, but you didn't note it in methods. Why you use consensus sequence of 16S rRNA? Is it just mistake of titles?

You work with one consensus sequence of ITS, but in previously articles Dr. Jeffrey R. Johansen and his co-authors (see descriptions Myxacorys, and Trichotorquatus) pointed than ITS trees for different operons shown some variation.

Could you discuss your previously data "percent dissimilarities >4% generally indicate that strains belong to different species" (Pietrasiak et al., 2021). What about dissimilarity of 16S-23S ITS sequences among members of Copelandiella more than 6.9%?

Author Response

  1. You recognized the position of Leptothermofonsia as the member of Kovacikia genus and I agree that that clade is monophyletic, but could you add some sequences to your trees to estimate a position of following strains KR137595, KR137599, KR137576, KY924323, DQ431005? Also, in my opinion, you able to consider the lineage of Leptolyngbya crispata AY239599, AY239598 and your described Leptolyngbya species.
    Thank you, we have added these strains to the analyses. However, there were no significant changes. It required an extra week of analyses to replace all impacted figures and tables, so perhaps the reviewer should consider these types of requests more carefully in the future.  The CENA strains and Greenland 10 were not even identified to species or in some cases not identified to genera.  The SEV clade of Leptolyngbya was already represented before adding these addition L. crispata strains. Seven additional sequences will have almost no impact on a phylogeny with more than 300 sequences.
    2. Why didn't you use recently Nodosilinea taxa in your analysis? I mean, new species cited in
    Davydov D., Shalygin S., Vilnet A. New cyanobacterium Nodosilinea svalbardensis sp. nov. (Prochlorotrichaceae, Synechococcales) isolated from alluvium in Mimer river valley of the Svalbard archipelago // Phytotaxa, 2020. Vol. 442(2). P. 61-79. DOI: https://doi.org/10.11646/phytotaxa.442.2.2
    Cai, F.; Li, S.; Zhang, H.; Yu, G.; Li, R. Nodosilinea hunanesis sp. nov. (Prochlorotrichaceae, Synechococcales) from a Freshwater Pond in China Based on a Polyphasic Approach. Diversity 2022, 14, 364. https:// doi.org/10.3390/d14050364
    Perhaps, your topology will change. 
    Thank you for your comment. We added N. svalbardensis to the analysis.  However, the 16S rRNA sequence for N. hunanensis is problematic.  There is a long read in the middle that has very divergent sequence, and when checked to see if it maintains secondary structure in the conserved helices of the 16S rRNA molecule, it does not.  We concluded that this sequence was poor quality and did not use it.  However, the ITS sequence of N. hunanensis appeared to be good, so we did use it in tht ITS analysis. There were no changes in the tree topologies.
    3. Please, add sequence similarity comparison of the 16S rRNA gene between all the Nodosilinea species.
    Percent similarity of 16S rRNA gene data is not very informative for species separation, so we have provided a table of ITS percent dissimilarity of all Nodosilinea species that have this sequence available.  It is Supplemental Table S3, as it is too large to put in the main text.
    4. In figures 2-4 you indicated consensus sequences, but you didn't note it in methods. Why you use consensus sequence of 16S rRNA? Is it just mistake of titles? You work with one consensus sequence of ITS, but in previously articles Dr. Jeffrey R. Johansen and his co-authors (see descriptions Myxacorys, and Trichotorquatus) pointed than ITS trees for different operons shown some variation.
    We have added more of an explanation of how we build consensus sequences in the methods.  The vast majority of clones we obtained had two tRNA genes in the ITS, and only in one case (Albertania prattii, see Table 6) did we find two different operons containing both tRNA genes.  We are very conscious of the operon problem, and took care not to mix orthologues.  In the case of A. prattii we included these two orthologues as we could not identify multiple operons in the other taxa and we wanted to show the degree of difference between the two orthologues.  They differ primarily in the V2 region and the spacer between tRNA-Ala and the Box-B helix.

5. Could you discuss your previously data "percent dissimilarities >4% generally indicate that strains belong to different species" (Pietrasiak et al., 2021). What about dissimilarity of 16S-23S ITS sequences among members of Copelandiella more than 6.9%?
We discuss this problem in TAXONOMIC NOTES for Copelandiella yellowstonensis.  The 16S rRNA sequences among our strains were very similar (>99.6%), the level of similarity which we generally find in the same species.  We could not exclude the possibility that the high dissimilarity in the ITS regions of these strains was due to multiple operons.  We took a conservative approach and put them all in the same species.  With more sampling we may end up dividing this species into additional species.  It appears there could be three species if there is not a problem with multiple operons.  

Reviewer 2 Report

Excellent work.

This is a well-conducted and written study that examines the diversity of simple trichal cyanobacteria in hot springs using a novel approach. A new genus with two species and eight other new species are described. Considering that the identification of cyanobacteria is problematic, this study makes an important contribution to the proper evaluation of cyanobacteria in general. Moreover, hot springs are extreme habitats that exist all over the world, so this study is of great importance.

Author Response

Thank you for your complimentary review of our article.

Authors

Reviewer 3 Report

Kastovsky et al. propose many new cyanobacterial species and one genus from the Yellowstone Park hot springs. This is a unique study. Although there are +7000 papers on Google Scholar involved in the cyanobacteria of Yellowstone Park, researchers were mostly focused on the whole microbial communities or Thermostichus (which they call Synechococcus). Now, we know the diversity better. I have some minor points below.

L13. Avoid words like ‘total’. A lot of evidence could be added to the paper.

L18-20. This sounds like a discussion point. If this is not a result of your paper, remove it.

The introduction should have a paragraph on the genera you investigated – Albertania, Kovacikia, Leptolyngbya, Nodosilinea, and Ocullatela. It will provide a reader background on the taxonomic status of these taxa.

I appreciate an introduction to hot springs in general. However, you are missing an introduction to the features of the Yellowstone hot springs you studied. At least some basic ecological characteristics. 

L96. Mobio does not produce the kit anymore, they have been sold by Quiagen for several years, or you have some old reserves

L119-120. Please add information on the similarity of the operons within one strain. 

Please store all alignments on the figshare or similar repository so that they are available to the reader.

The similarity table could be moved to supplements. It takes a lot of space.

L444 and elsewhere. Thresholds for the similarity values may differ. State, based on which paper, you use the ITS threshold.

Figure 15 and other trees. Highlight studied strains for a better orientation. 

L603 and elsewhere. What do you mean by slight? The differences should be presented as specifically as possible. 

L757. If the name Leptothermofonsia is not valid, I think you can easily include it in the Kovacikia with emended genus description. I can see only one issue. The clade of Kovacikia has low support.  Thus, I agree that it to early to make the revision. Perhaps, with the genomes, the situation will be cleared.

L844. You have to fill the data availability statement with links to the alignments repositories and where you find accession numbers of the sequences. 

One note. Although some people recently advocated that naming taxa in honor of people is not good, I disagree, and I am glad that you proposed names honoring scientists. 

Author Response

1. L13. Avoid words like ‘total’. A lot of evidence could be added to the paper.
Thank you, we changed it – to polyphasic approach. Total evidence approach is the term used among taxonomists dealing with non-algal taxa, and it means using all the evidence available, not all the evidence potentially available.  However, the reviewer is correct in knowing that cyanobacteriologists show a prefernce for using the term polyphasic approach.

2. L18-20. This sounds like a discussion point. If this is not a result of your paper, remove it.
These claims are results and conclusions from our study, so we propose to keep them in the abstract. We feel it is appropriate for an abstract to have the important findings and conclusions.

3. The introduction should have a paragraph on the genera you investigated – Albertania, Kovacikia, Leptolyngbya, Nodosilinea, and Ocullatela. It will provide a reader background on the taxonomic status of these taxa.
We feel that these are well known genera and that it would take much more than a paragraph to cover.  However, we have added a sentence at the end of the introduction introducing the genera with which we are dealing.
4. I appreciate an introduction to hot springs in general. However, you are missing an introduction to the features of the Yellowstone hot springs you studied. At least some basic ecological characteristics. 
Done, thank you
5. L96. Mobio does not produce the kit anymore, they have been sold by Quiagen for several years, or you have some old reserves
I believe we did have reserves for some time, but likely we changed over before this study, so we cite the Qiagen kit we now use.

6. L119-120. Please add information on the similarity of the operons within one strain. 
We increase our description of how we obtained consensus sequences.  Operons had to be near identical in ITS regions to consider them orthologous and available for making consensus sequences.  There was only one instance in the study where we obtained multiple multiple operons in a strain, Albertania prattii, and we point this out in Table 6 where the dissimilarity can be seen to be quite high (>16%).

7. Please store all alignments on the figshare or similar repository so that they are available to the reader.
We apologize, but we will not agree completely with this comment. We have deposited a file with all taxon names and GenBank accession numbers, and include the DOI for these data in response to this request (see Methods, Phylogenetic Analyses).. All input data (GenBank accession number, uncollapsed tree in supplementary material) and description of the methodology are available to the reader. We consider our alignment to be an original and somewhat proprietary result and do not have a good experience with publishing it, given the easy potential for this data to be plagiarized.  The work is repeatable given that all accession numbers are available and the phylogenetic methods are clearly stated.

8. The similarity table could be moved to supplements. It takes a lot of space.
Thank you, they really do occupy a lot of space. However, the information in them is crucial for our conclusions. We therefore suggest (see new version of the text) to move the large tables on Leptolyngbya and Nodosilinea to Supplementary Material. We propose to leave the significantly smaller tables concerning the other genera in the main tetx. We would like to ask the editor for a final decision on this issue.

9. L444 and elsewhere. Thresholds for the similarity values may differ. State, based on which paper, you use the ITS threshold. 
These have been added the first time the thresholds are mentioned, headings for Tables 1 and 2.

10. Figure 15 and other trees. Highlight studied strains for a better orientation. 
We have marked them with red font in both phylogenies and the tables.

11. L603 and elsewhere. What do you mean by slight? The differences should be presented as specifically as possible. 
We have been more specific about what we mean by slightly, thank you.

12. L757. If the name Leptothermofonsia is not valid, I think you can easily include it in the Kovacikia with emended genus description. I can see only one issue. The clade of Kovacikia has low support.  Thus, I agree that it to early to make the revision. Perhaps, with the genomes, the situation will be cleared.
Another reviewer asked that we add additional sequences, these did not change topology, but the support for the Kovacikia clade is now much better (1.0 in posterior probability).  Additional taxon sampling will likely improve the suppport in future ML trees

13. L844. You have to fill the data availability statement with links to the alignments repositories and where you find accession numbers of the sequences. 
Figshare DOI number added.

14. One note. Although some people recently advocated that naming taxa in honor of people is not good, I disagree, and I am glad that you proposed names honoring scientists.
We agree, plus the work of the chosen scientists is closely related to thermal springs in general and Yellowstone in particular.

Round 2

Reviewer 1 Report

Thank you, for your answers and changes in the manuscript. Unfortunately I can't see supplementary tables in the submission.

Author Response

Dear Editor and Reviewers,
I've re-uploaded our supplementary tables into the editorial system. I hope it will be fine now.

Best regards,

Jan Kaštovský